# 3D chromatin interactions involving *Drosophila* insulators are infrequent but preferential and arise before TADs and transcription

Olivier Messina [1], Flavien Raynal [2], Julian Gurgo[1], Jean-Bernard Fiche[1], Vera Pancaldi [2,3] ✉ & Marcelo Nollmann [1] ✉

In mammals, insulators contribute to the regulation of loop extrusion to organize chromatin into topologically associating domains. In *Drosophila* the role of insulators in 3D genome organization is, however, under current debate. Here, we addressed this question by combining bioinformatics analysis and multiplexed chromatin imaging. We describe a class of *Drosophila* insulators enriched at regions forming preferential chromatin interactions genome-wide. Notably, most of these 3D interactions do not involve TAD borders. Multiplexed imaging shows that these interactions occur infrequently, and only rarely involve multiple genomic regions coalescing together in space in single cells. Finally, we show that non-border preferential 3D interactions enriched in this class of insulators are present before TADs and transcription during *Drosophila* development. Our results are inconsistent with insulators forming stable hubs in single cells, and instead suggest that they fine-tune existing 3D chromatin interactions, providing an additional regulatory layer for transcriptional regulation.

Eukaryotic chromosomes are organized in a multi-layered structure comprising chromosome territories, compartments, topologically associating domains (TADs) and nano-domains[1,2]. Notably, this multi-scale organization of the genome is conserved from *Drosophila* to mammals[3–6]. However, the mechanisms responsible for the acquisition and maintenance of these structures seem to differ between species.

In vertebrates, TADs are often separated from each other by convergent CCCTC-binding factor (CTCF) sites localized at TAD boundaries. TAD borders bound by CTCF/cohesin form "focal chromatin loops" in contact matrices[7]. These specific looping interactions may facilitate the communication between genes and their cis-regulatory elements (CREs, e.g. enhancers and promoters) most often localized within TADs[8,9]. In mammals, the formation of TADs is

thought to involve loop extrusion, a mechanism by which Structure Maintenance of Chromosome (SMC) proteins (e.g. cohesin) bind chromatin and reel it in until they encounter properly-oriented CTCF sites[10]. In contrast, the *Drosophila* homolog of CTCF (dCTCF) binds preferentially within TADs and is only mildly enriched at TAD borders[4,6] with no preferential convergent orientation as in mammals[11]. These results suggest that other mechanisms may instead be responsible for the establishment of TADs in *Drosophila*.

While CTCF is the main chromatin insulator in mammals, *Drosophila* contains tens of non-evolutionary conserved insulator binding proteins (hereafter IBPs)[12]. Since before the genomic era, *Drosophila* insulators were shown to be involved in the regulation of long-range chromatin interactions: either by blocking enhancer-promoter

[1]Centre de Biologie Structurale, Univ Montpellier, CNRS UMR 5048, INSERM U1054, 34090 Montpellier, France. [2]CRCT, Université de Toulouse, Inserm, CNRS, Université Toulouse III-Paul Sabatier, Centre de Recherches en Cancérologie de Toulouse, Toulouse, France. [3]Barcelona Supercomputing Center, Barcelona, Spain. ✉e-mail: vera.pancaldi@inserm.fr; marcelo.nollmann@cbs.cnrs.fr

interactions or by establishing barriers between chromatin states[13–15]. Early genome-wide studies showed that insulators preferentially bind to genomic regions containing housekeeping genes and highly transcribed regions[16]. In addition, IBPs frequently bind to TADs borders[4,6,17–19] that can often interact in 3D[20]. Taken together, these data suggest that insulators may be involved in the organization of *Drosophila* TADs.

Recent studies suggest different modes of action. On one hand, insulators may promote TAD border interactions by forming contacts between insulator factors[21–23]. On the other hand, insulators may not form CTCF-like focal chromatin loops, but rather restrict interactions between domains[24].

Here, we investigated the role of insulators in the 3D organization of the *Drosophila* genome by combining advanced bioinformatics analysis and Hi-M, an imaging-based method we recently developed[25] to detect the 3D positions of multiple genomic loci in single cells[25]. First, we show that genomic regions occupied by insulators display preferential interactions genome-wide. These preferred interactions occur inside TADs and can also span TAD borders. Second, we show that chromatin regions displaying the most prominent 3D interactions are preferentially bound by insulators. We detect TAD border preferential interactions, but these represent the minority of the interactions detected. Interestingly, non-border interactions quantitatively increased with the occupancy of IBPs. By visualizing 3D chromatin structure at the *dpp* locus, we observed, however, that spatial colocalization between insulators is infrequent and similar to neighboring regions not bound by insulators. Finally, by mapping preferential interactions during development, we found that non-border regions harboring insulators display a tendency to preferentially interact before the emergence of TADs and transcription.

## Results

### Genomic regions displaying preferential interactions are predominantly bound by chromatin insulators

To shed light onto the roles of *Drosophila* insulators in 3D genome organization during early embryogenesis, we deployed Chromatin Assortativity analysis (ChAs)[26,27]. Assortativity measures the preference for the nodes of a network to interact with other nodes that have the same characteristics. In ChAs analysis, a chromatin interaction network is built from a genome-wide contact map[28]. This network represents the genomic loci (nodes) displaying high frequency interactions amongst each other (Fig. 1a, see "Chromatin assortativity" in "Methods"). Chromatin assortativity for a given factor is calculated by estimating whether nodes bound by this factor interact with other nodes with the same factor more than expected at random. Thus, a factor with positive assortativity is enriched in chromatin loci that preferentially interact.

We applied ChAs analysis to study chromatin organization of *Drosophila* embryos at nuclear cycle 14 (nc14)[20], a developmental stage coinciding with the zygotic genome activation (ZGA) and with the emergence of TADs[20]. For this, we obtained chromatin interaction networks by mapping preferentially interacting chromatin regions using Chromosight[29] on Hi-C data (Figs. S1a–d). Remarkably, the constructed network exhibits high overlap with previously annotated loops in the *Drosophila* embryo (Fig. S1a)[30]. Chromosight detects preferential chromatin interactions by segmenting the genomic regions displaying local maxima in the observed/expected Hi-C map. In mammals, loops often appear as clear focal peaks[7], however most of the Chromosight-annotated interactions from nc14 Hi-C data do not appear as focal peaks in the observed Hi-C map (Fig. S1d). This is consistent with many preferential contacts in *Drosophila* representing low-frequency interactions. Next, we annotated these chromatin networks with the binding patterns of publicly available ChIP-seq datasets (features, Fig. 1a) and calculated chromatin assortativities for a wide panel of chromatin binding factors, including insulator and insulator-

associated proteins (BEAF-32, CBP, CHRO, CP190, dCTCF, DREF, FS(1) h, GAF, L(3)MBT, Pita, Mod(mdg4), Su(HW), Z4, ZIPIC and Zw5), pioneering factors (Zelda), RNA polymerase II (RNAPII CTD phospho-Ser5: S5P), Polycomb group proteins (Pc, Ph) and the cohesin subunit (Rad21).

Chromatin assortativity Z-scores (hereafter ChAs Z-Scores) are calculated to estimate if ChAs for a feature is higher than expected for regions separated by similar genomic distances, indicating the importance of 3D interactions for establishing preferential contacts. Regions enriched in Zelda, Polycomb group proteins (Pc and Ph), and RNAPII CTD phospho-Ser5 (S5P) displayed positive ChAs Z-scores (Fig. S1e), consistent with previous findings[31–33]. In contrast, ChAs Z-Scores were highly variable between IBPs (Fig. 1b), indicating that different insulators may contribute unequally to the formation of preferential contacts. A sub-group of IBPs displayed high assortativities (ChAs ZScore > 2), including the insulator and insulator-associated proteins: BEAF-32, CHRO, DREF, L(3)MBT, Pita, Z4, ZIPIC and Zw5 (Fig. 1b). Notably, cohesin (Rad21), dCTCF, and a second sub-group of IBPs including CBP, CP190, Fs(1)h, GAF, Mod(mdg4) and SU(HW) displayed low assortativity and low Z-Scores (ChAs Z-Score <2, Figs. 1b and S1e). To validate the robustness of these results, we performed similar analysis for different sets of Chromosight parameters (see "Chromatin assortativity" in "Methods") generating larger networks that include lower-frequency interactions. ChAs Z-Scores were highly correlated between networks, and the insulator factors exhibiting the highest ChAs Z-scores were the same independently of the network size or loop size distribution (Fig. S1f–h). For some insulators the ChAs Z-Score increase was larger than proportional in the networks including longer-range contacts (e.g. GAF), while for others the ChAs Z-Score increased less than proportionally (e.g. Fs1h, CTCF). This is consistent with these factors being slightly more/less assortative depending on the network loop size distribution. We note, however, that these factors still displayed the lowest assortativities in all networks.

Low assortativity scores can arise when the presence of a factor is not associated with a preferential interaction (Fig. 1a), or if the factor is present either in a very small or in a very large proportion of them. For instance, GAF is often bound to the anchors of focal loops clearly visible in Hi-C and micro-C datasets[30,31,33]. These focal loops, however, represent a small proportion of preferential interactions in our network (~11%, Fig. S1a), consistent with the low ChAs Z-Scores we observed. We note that GAF binds to thousands of sites genome-wide (3842), however only a small fraction of these sites correspond to focal loop anchors (<620)[30]. Taken together, these results are consistent with only a small number of GAF binding peaks being involved in focal loops and in regulating transcriptional activation and repression[30,31,33].

### Insulator binding increases the strength of preferential chromatin contacts

Next, we complemented ChAs with aggregation peak analysis (APA)[7]. This method relies on the calculation of pairwise, intra-arm autosomal contact frequencies between genomic regions bound by a given factor (i.e. peak) (Fig. S1i). The statistical relevance of these contacts is estimated by calculating the average of the log2(Observed/Expected) distribution of the Hi-C signal at all peak regions (see "Log2(O/E) and Hi-C aggregate plot analysis" in "Methods"). Thus, the log2(O/E) ratio is positive when contacts occur at frequencies higher than expected and is negative when contact frequencies are lower than expected for regions separated by the same genomic distance.

Notably, the positive correlation between ChAs and log2(O/E) (Fig. 1c, Supplementary Data 1) indicates that factors displaying high assortativities are bound to chromatin regions that exhibit the most preferential interactions. Remarkably, most of the insulator factors displaying positive ChAs Z-Scores also exhibited positive log2(O/E) (BEAF-32, CHRO, DREF, Z4, ZIPIC and Zw5) (hereafter referred to as Class I insulators) (Figs. 1b and S1j). The peaks observed for negative

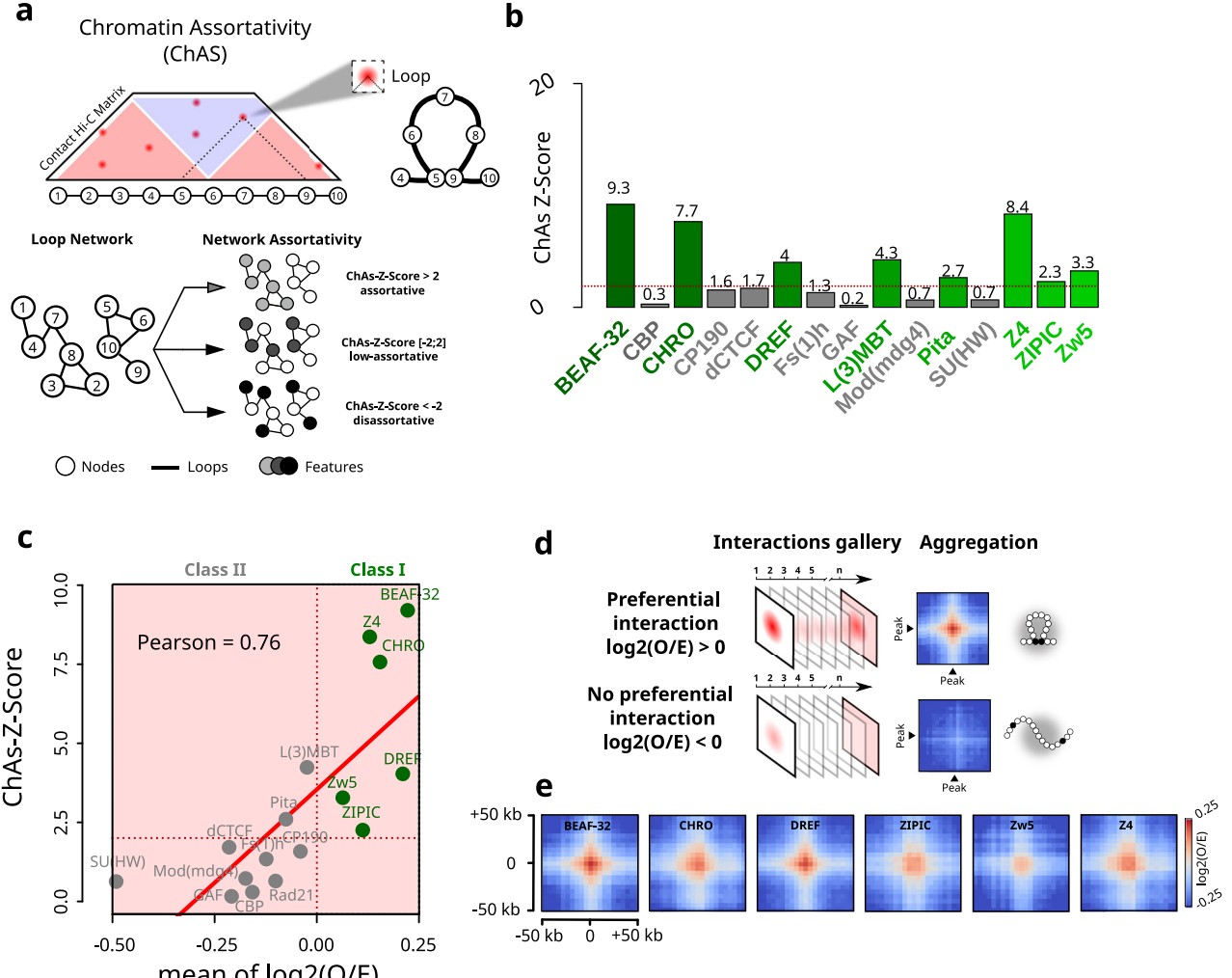

**Fig. 1 | Genomic regions displaying preferential interactions are predominantly bound by chromatin insulators. a** Cartoon illustrating Chromatin Assortativity (ChAS) of chromatin binding factors in a network of chromatin contacts. 5-kb genomic bins are represented by nodes in the chromatin network. Nodes are connected to each other if they form loops in Hi-C data, called by Chromosight[29]. Nodes are color-coded according to the presence or absence of a given chromatin binding factor (features). The assortativity is then calculated for each different factor (see "Methods"). **b** Bar plot illustrating ChAS Z-Scores for 15 IBPs in the nc14 chromatin network classified by alphabetical order. The horizontal red dashed line represents the ChAS Z-Score = 2 threshold considered in this study. **c** Pearson's correlation between ChAS Z-Scores from ChAS (y) and mean of log2(O/E) from APA (x) for the 15 IBPs tested. Class I and II are delineated by a vertical red dashed line centered at log2(O/E) = 0. **d** Cartoon illustrating the Hi-C aggregate map procedure around pairs of specific chromatin binding factors (peak). The first line illustrates the log2(O/E) aggregate map expected for a given factor involved in preferential contact formation and the second for a factor not involved in preferential contact formation (see "Methods"). **e** Aggregate Hi-C plots of Class I IBPs regions in nc14 embryos[20]. Maps show the log2(O/E) in a 50 kb window around the crossing point of two Class I IBPs regions: BEAF-32, CHRO, DREF, ZIPIC, Zw5, Z4. Source data are provided as a Source Data file.

log2(O/E) values (referred to as *peak 2* in Fig. S1j) are related to longer-range contacts. Consequently, it can be inferred that Class I insulator sites exhibit a higher tendency to interact with each other at shorter distances (<250 kb, Fig. S1k, l). Thus, Class I insulators occupy genomic regions displaying the most preferential interactions, and conversely, the genomic regions they occupy tend to preferentially interact in 3D in nc14 embryos.

Next, we investigated the specificity of preferential chromatin interactions by using Hi-C aggregate plot analysis[20] (Fig. 1d, see "Log2(O/E) and Hi-C aggregate plot analysis" in "Methods"). Class I IBPs displayed a well-defined center spot, indicating that presence of IBPs at both loop anchors reinforces preferential 3D interactions (Fig. 1e). Similar results were observed for Zelda and RNAPII (Fig. S1m). In contrast, factors with low assortativity and/or negative log2(O/E) did not exhibit centered spots (Fig. S1m, n), likely due to positive and negative log2(O/E) values for different regions averaging out.

Preferential interactions captured by Chromosight are highly variable and often do not appear as focal peaks (Fig. S1d). We further

analyzed the impact of this variability in our analysis by focusing on BEAF-32 –the insulator displaying the highest log2(O/E) ratio and ChAS Z-Score– and investigated how the interaction preference depended on the number of peaks aggregated. For this, we first calculated the distribution of log2(O/E) values for different numbers of BEAF-32 peaks averaged using bootstrapping (Fig. S1o, left panel, see "Log2(O/E) and Hi-C aggregate plot analysis" in "Methods"). On average, most of the 2- and 5-peak aggregations displayed low or no preference. Nonetheless, most aggregations exhibited positive log(O/E) values when 25 or more BEAF-32-bound regions were averaged. Overall, these results indicate that interactions between different BEAF-32 anchors are highly variable and often display low or no preference. In support of these conclusions, well-centered peaks in Hi-C aggregate analysis were observed only after a sufficient number of BEAF-32-bound regions were aggregated (Fig. S1o, right panel). All in all, these analyses agree with our previous observations (Fig. S1d), and suggest that interactions between insulator-bound genomic regions are on average preferential, but highly variable and often weak.

To investigate whether IBPs act together to promote preferential chromatin interactions, we employed Cross-ChAs and AND-ChAs[27]. Cross-ChAs measures assortativity of two different proteins, giving information about frequency of interactions joining fragments with one protein on either side. Instead, AND-ChAs measures assortativity of two different proteins considering that connected nodes are bound by a pair of factors, and therefore provides information about interaction frequencies of co-occupied regions. We computed Cross-ChAs and AND-ChAs Z-Scores for each pair of factors investigated previously (Fig. S1p, q). Cross-ChAs shows that class I insulators (BEAF-32, Chromator, Z4, PolII, Zelda, L3(MBT), DREF) tend to display high cross-assortativities, suggesting that anchors bound by either of these factors tend to preferentially interact. AND-ChAs shows that DNA fragments containing colocalized class I insulators (BEAF-32, Chromator, Z4, PolII, Zelda, L3(MBT), ZIPIC) interact preferentially with each other. Thus, pairs of class I IBPs can be found at each anchor of strong loops. These results are consistent with Class I IBPs often interacting together to promote formation of preferential chromatin contacts in nc14 embryos.

## Most insulator-bound preferential contacts involve non-border chromatin regions

TAD borders in *Drosophila* are mostly occupied by insulators[4,6], with only 4% of borders lacking insulator sequences[19]. In *Drosophila*, ensemble analysis showed that adjacent TAD borders tend to preferentially interact[20], however contact maps do not display focal chromatin peaks as those observed in mammals[7,34,35]. To determine whether our unbiased network analysis was able to recover preferential interactions between TAD borders, we calculated whether TAD borders were assortative in the network of chromatin interactions generated by Chromosight for nc14 embryos. This analysis shows that TAD borders appear highly connected to each other in the interaction network (Figs. 2a and S2a). This connectivity results in considerably higher ChAs values for TADs borders (Fig. 2b, blue dashed line) as compared to randomized networks (Fig. 2b, black distribution). As expected, and consistent with previous analysis[20], TAD borders exhibited a well-centered interaction spot in Hi-C aggregate plots (Fig. S2b).

The strength of a TAD border, as assessed by its insulation score (IS), is positively correlated to the binding level of insulator proteins[19,36]. Given this correlation, we tested if the presence of IBPs at TAD borders is also associated with their interaction preference by stratifying TAD borders into five equally-sized categories according to their IS and by computing Hi-C aggregate plots for each category. Notably, the level of preferential interactions between TAD borders increased with insulation strength (Fig. 2c), providing indirect evidence for a role of IBPs in contributing to TAD border interactions.

Next, we wondered whether preferential chromatin contacts may be detected in locations other than TAD boundaries and what their determinants may be. To this aim, we divided the interactions in our chromatin network into three categories: border/border (blue), border/non-border (red), and non-border/non-border (black) (Fig. 2d). Then, we quantified the occurrence of each type of interaction by quantifying the presence of a border on each loop anchor. Notably, preferential interactions involved a border in one or both of their anchors in a minority of cases (<1% for border/border and ~6% for border/non-border), with the overwhelming majority of preferential interactions involving non-borders (>93%) (Fig. 2e).

To better understand the role of insulators in each of these interaction categories (i.e. border/border, border/non-border and non-border/non-border), for each category we calculated the proportion of interactions displaying binding of Class I IBPs in two, one or none of the anchors. The vast majority of the anchors for all categories were bound by at least one Class I IBPs (>92%, Fig. 2e, right pie charts). Anchors in border–border interactions are most often bound by two Class I IBPs (~90%), and in a smaller proportion by a single class I IBP. This trend was similar for the other categories, further supporting a

role of class I IBPs in the mediation of chromatin loops that in most cases do not involve TAD borders.

The number of loop anchors corresponding to TAD borders is considerably larger than the number of non-borders. Thus, we estimated the probability with which a border may take part in a loop by calculating the proportion of borders participating in loops (either in one or both anchors). We found that ~38% of borders take part in loops in our Chromosight network (Fig. S2c), with the majority of them participating as a single anchor (~36.8%) (Fig. S2d). Next, we calculated similar statistics for Class I IBPs non overlapping with borders. Notably, we found that the propensity of non-border IBP peaks to form loops was always lower than that of TAD borders (Fig. S2c, d). Overall, these results are consistent with Class I IBPs binding at loci displaying preferential looping, at both border and non-border regions.

To further support this conclusion, we performed aggregation Hi-C analysis on non-border/non-border regions occupied by Class I IBPs. Notably, this analysis displays a clear peak (Fig. 2f), suggesting preferential interactions between anchors containing Class I IBP sites. Consistently, interactions mediated by Class I IBPs at non-border regions increased with ChIP intensity (binding strength) (Figs. 2g and S2e) of both anchors (Figs. 2h and S2f). All in all, these analyses suggest that class I insulators participate in mediating preferential interactions between border and non-border chromatin regions. These analyses, however, do not inform us on how frequently these preferential interactions occur in single cells, or whether they involve two or multiple anchors.

## Insulator-bound chromatin regions only infrequently co-localize in 3D

Sequencing-based 3C methods only provide relative interaction frequencies, thus we turned to DNA-FISH, a technique that can quantify absolute physical proximity frequencies[37,38]. As conventional DNA-FISH can only measure proximity between a limited number of genomic targets, we used Hi-M, a multiplexed imaging method that enables the detection of tens of genomic loci at once[25,32,39] (Fig. 3a). Specifically, we imaged the 3D chromatin organization of the *dpp* locus (*chr2L: 2343645-2758688* dm6) in intact nc14 *Drosophila* embryos at ~12 kb resolution. The *dpp* locus contains three TADs and several regions displaying high levels of class I insulator binding, named barcode I1 to I10 (Figs. 3b and S3a). To cover this locus, we designed 34 equally-spaced barcodes that label insulator-bound and insulator-free genomic regions (Fig. 3c). Nuclei and barcodes were registered, segmented and localized as in previous studies[25,32] (see "Image processing" in "Methods"), with similar barcode detection efficiencies (Fig. S3b–d). Ensemble pairwise distance maps were built by calculating the median of the full pairwise distance (PWD) distributions (Fig. S3e). Proximity maps were constructed by calculating the frequency of colocalization for each pair of barcodes from chromatin traces (Fig. S3f) using a pre-established distance threshold that maximizes the correlation between Hi-M and Hi-C datasets ($d = 200$ nm, Fig. S3g, h), and that was previously used for similar studies[32]. The number of traces acquired was sufficient to ensure a statistically representative ensemble map (Fig. S3i, see "Image processing" in "Methods").

The proximity and PWD distance maps revealed multiple regions displaying preferential 3D spatial proximity (Fig. 3c). These mostly corresponded to the TADs called from Hi-C data (Fig. 3b, c, blue arrows) and from Hi-M proximity frequency maps (Fig. 3c, insulation score, and domainogram). We note that TAD3 is more insulated than the other two TADs in this region, and that it is flanked by multiple IBP peaks. This is consistent with the role of IBPs in TAD insulation. To quantify the frequency at which insulator-bound regions spatially colocalized in a population of single cells, we calculated the cumulative average proximity frequencies between insulator-bound regions and control regions for different cutoff distances (Fig. S3j). At the cutoff distance used to calculate proximity maps (200 nm), insulator

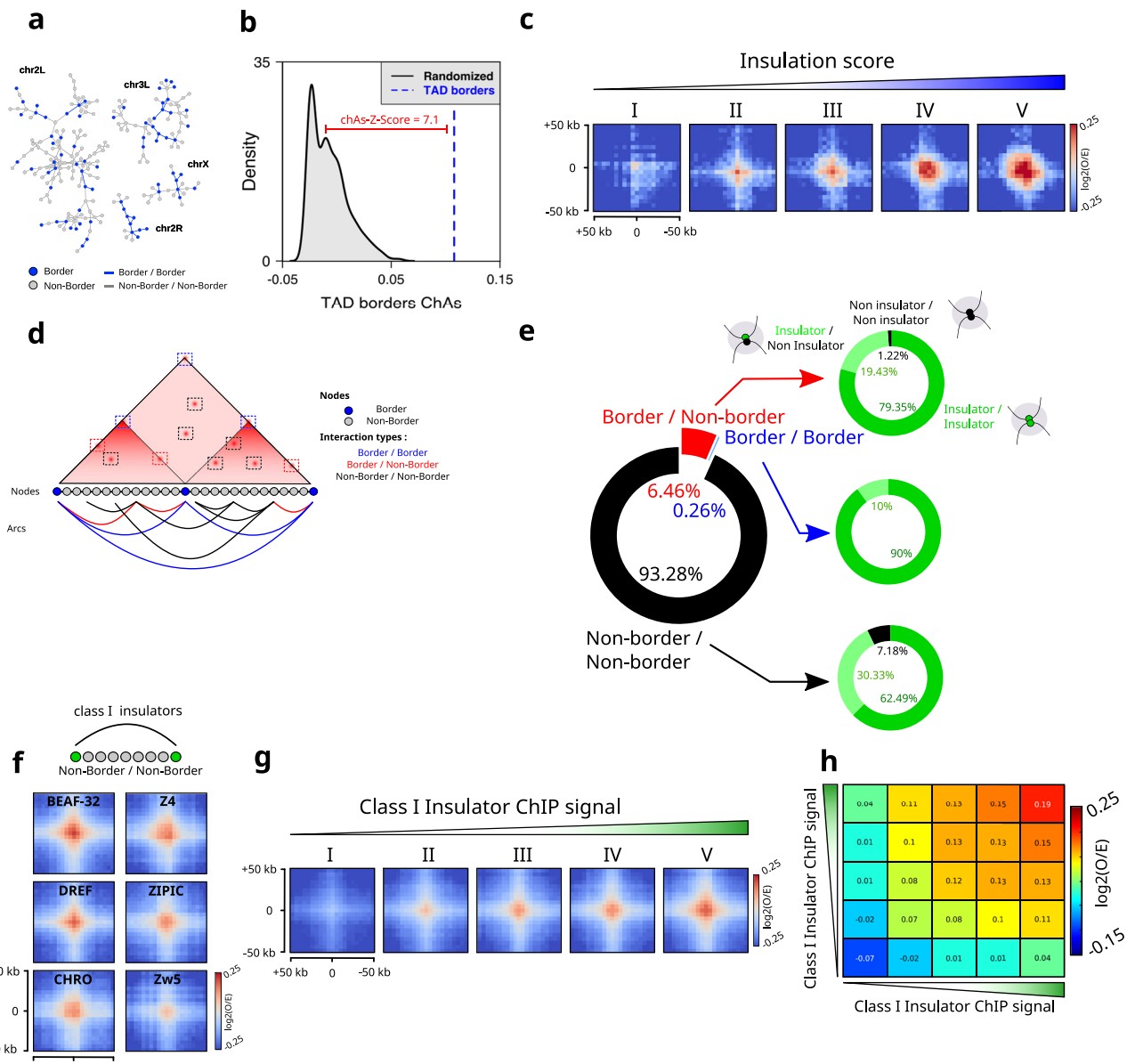

**Fig. 2 | Border–border and non-border interactions are favored by an increase in insulation score and an increase in IBPs binding, respectively. a** Chromosight chromatin subnetworks from Hi-C data in nc14 embryos[20]. Each node of the network is a chromatin fragment, blue nodes represent nodes where a TAD boundary is found, and edges represent significant 3D interactions. **b** ChAs Z-Score for TAD borders (blue) versus distribution of ChAs scores for randomized networks (black). The ChAs Z-Score is calculated for TAD borders based on comparing ChAs values with the distribution of the ChAs in randomized networks (see "Methods"). **c** Aggregation Hi-C plots for TADs borders in nc14 embryos stratified into five equal-size category groups (I, II, III, IV and V) with an increasing level of insulation score. **d** Cartoon illustrating the different types of interaction observed in Hi-C dataset. Genomic bins are represented by color-coded nodes. Arcs represent interactions between pairs of genomic bins. Border/Border interactions are shown in blue, Border/Non-Border in red and non-border/non-border in black. **e** Donut chart representing the loop distribution called by Chromosight into the different types of interactions (left panel). Donut charts illustrating the quantification of Class I IBPs bound on each side of the anchored interaction (right panel). **f** Aggregation Hi-C plots for non-border regions bound by Class I IBPs in nc14 embryos. **g** Aggregation Hi-C plots for Class I IBPs in nc14 stratified into five equal-size category groups (I, II, III, IV and V) with an increasing ChIP signal. **h** Log2(O/E) average interaction frequencies between five categories of Class I IBPs regions ranked by increasing ChIP signal in nc14 embryos. Source data are provided as a Source Data file.

barcodes co-localized on average only in a small fraction of cells (~12.19%, Fig. S3j, green curve and inset). As expected, the proximity frequency monotonously increased with cutoff distance, but remained low for cutoff distance thresholds used in this and other studies (<200 nm)[25,32,40]. Thus, we conclude that the average colocalization between insulator-bound regions within and between TADs is rather infrequent, consistent with colocalization of insulator barcodes occurring only in a small proportion of cells (i.e. large cell-to-cell heterogeneity) or/and with colocalization being highly dynamic.

Next, we investigated the specificity of insulator barcode co-localizations by calculating the proximity frequency versus cutoff distance curve for non-insulator (control) barcodes located at similar genomic distances (Fig. S3j, black curve). For this, we averaged 10 sets of control barcodes. At a cutoff distance of 200 nm, control barcodes co-localized at similar frequencies than insulator barcodes (10.8% and 12.19%, respectively). Next, we calculated how proximity frequency depended on genomic distance for both insulator and control barcodes, using a fixed cutoff distance of 200 nm (Fig. 3d). This analysis

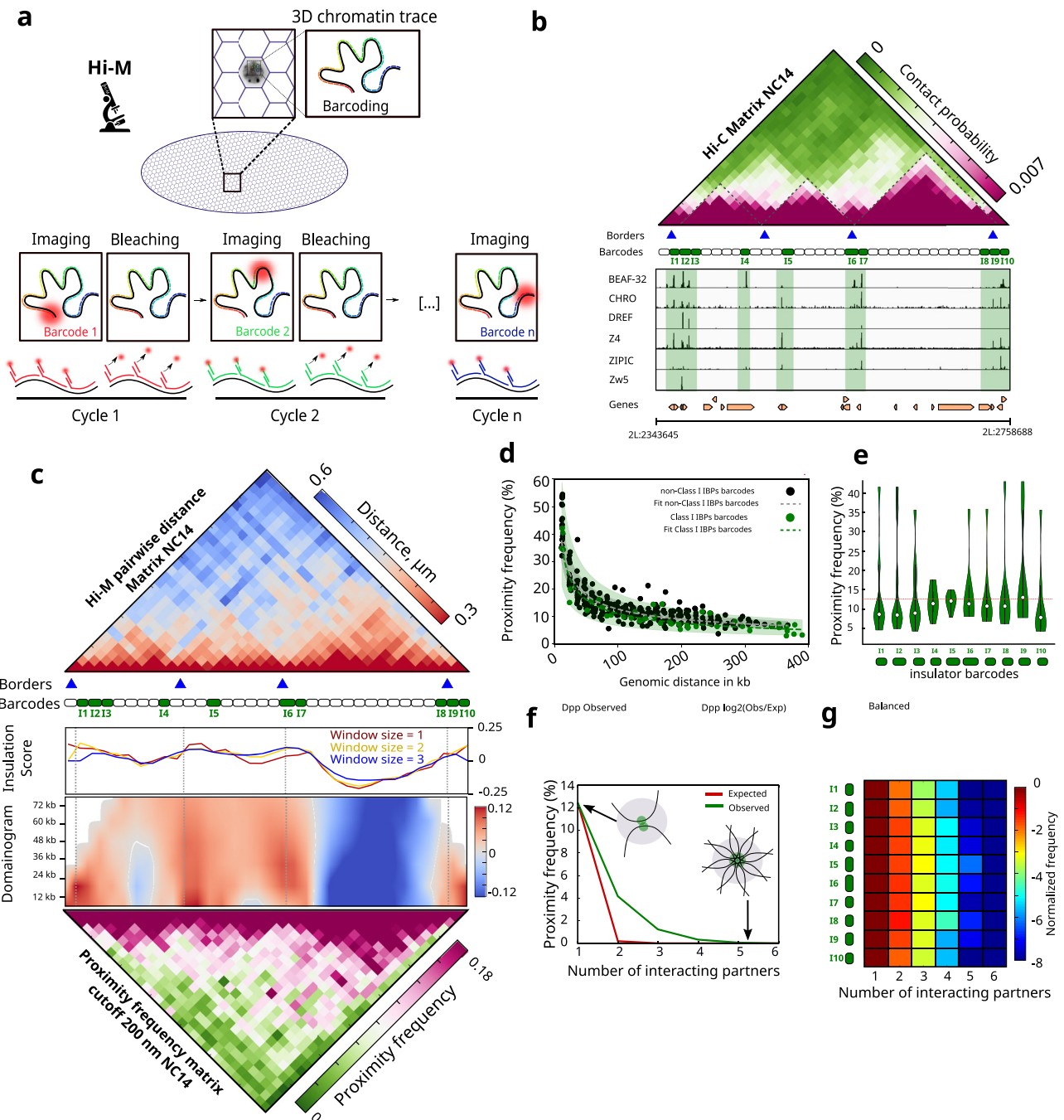

**Fig. 3 | Hi-M reveals visible interactions between insulator-bound chromatin regions.** **a** Cartoon illustrating the imaging-based strategy used to study chromosome conformation at the single cell level in intact *Drosophila melanogaster* embryos (Hi-M). **b** Top: nc14 Hi-C matrix along the *dpp* locus (2L:2343645-2758688) in *Drosophila melanogaster* (dm6). Purple and green represent high and low contact probabilities, respectively. Identified TADs borders from nc14 embryos[20] are represented by blue triangles. TADs are highlighted on the matrix with black dashed lines. Barcodes used for Hi-M sequential imaging are represented as boxes, with barcodes bound by Class I IBPs displayed in green. Bottom: ChIP-seq profiles for Class I IBPs (BEAF-32, Chromator, DREF, Z4, ZIPIC, Zw5) aligned with genomic coordinates and gene locations. **c** Top: Hi-M pairwise distance (PWD) matrix for nc14 embryos constructed from 23531 traces from 22 embryos. Red and blue represent low and high distances, respectively. Middle: insulation score derived from Hi-M data with different window sizes (1, 2 and 3 bins), and domainogram (see "Methods"). Bottom: proximity frequency matrix from nc14 embryos (cutoff

distance: 200 nm). Pink and green represent high and low proximity frequencies, respectively. **d** Scatter plot illustrating the dependence of proximity frequency with genomic distance (cutoff distance: 200 nm) for Class I IBPs barcodes (green) and for non-Class I IBPs barcodes (black). Dashed black and green lines at the center of the error bands represent polynomial fits for each distribution, along with 95% confidence intervals. **e** Violin plot distributions representing the frequency with which each insulator barcode interacts with each other insulator barcode in our oligopaint library for nc14 embryos (see "Methods"). The dashed red line represents the mean. **f** Observed and expected proximity frequency versus number of interacting partners for Class I IBPs barcodes for nc14 embryos (green and red, respectively). **g** Histograms of Class I IBPs preferential interaction as a function of the number of interacting partners normalized by the number of pairwise interactions for nc14 embryos. Insulator barcodes are indicated on the left (I1–I10). The color-scale represents the normalized frequency in log2-scale. Source data are provided as a Source Data file.

revealed that, at least at the *dpp* locus, barcodes co-localize at similar frequencies irrespective of whether they contain insulators. Proximity frequencies dropped with genomic distance, as expected, but the difference between insulator and non-insulator barcodes remained small for all genomic distances. We note that use of larger cutoff distances increases the proximity frequency, but this would happen for both insulator and non-insulator barcodes. Overall, these results show that insulators coalesce in space infrequently, and only at slightly higher frequencies than non-insulator regions.

### Insulator barcodes most frequently co-localize in pairs

The existence of multiple focal peaks in the Hi-M matrix can be explained by two different models. On one hand, a low fraction of single cells can form rosette-like structures where multiple insulator-bound regions come together in space at once, as suggested by previous models[41]. On the other hand, different combinations of insulator barcodes may co-localize at low frequencies in a pairwise manner in single cells. In this case, the multiplicity of peaks in the Hi-M matrix would arise from ensemble averaging. To discern between these two models, we calculated how often insulator barcodes were proximal (i.e. at a distance ≤ 200 nm) to any other insulator barcode in single cells. This frequency was comparable for all the insulator barcodes investigated, and on average lower than 12% (Fig. 3e). Thus, in single cells, insulator barcodes interact with any other (genomically close) insulator barcode at low frequency.

Finally, to explore if these rare spatial encounters involved multiple insulator-bound regions, we calculated the proportion of clusters containing two (i.e. pairwise cluster) or multiple insulator barcodes (multiway cluster). Clusters containing only two insulator targets were the most common in all cases (>65%) (Fig. 3f). Next, we calculated the frequency of multiway clusters as a function of the number of barcodes in a cluster for all barcodes combined (Fig. 3f) or for each barcode independently at a distance ≤200 nm (Fig. 3g) and for different distance thresholds (Fig. S3k). We note that at larger cutoff distances (e.g. 400 nm) multiple barcodes can frequently coalesce in space, but we don't consider these to represent multiway clusters because of the large distances involved. The frequency of multiway clusters rapidly decreased with the number of co-localizing targets but was only slightly higher than what would be expected by chance (Fig. 3f, see "Multiway proximity frequency analysis" in "Methods"). All in all, these results indicate that insulator-bound regions rarely form clusters with more than two insulators, and when they do, they contain only a very limited number of insulator-bound regions.

### Preferential interactions between class I insulators arise before TADs and transcription

Previous studies showed that preferential spatial proximity between *cis*-regulatory elements (e.g. enhancers and promoters) can occur before nc14[32], the nuclear cycle at which most zygotic genes get activated and when TADs first emerge[20]. We reasoned that insulators may display similar features. To test this hypothesis, we first analyzed the changes in accessibility of class I IBP sites at different time points within nc12–13[42]. Surprisingly, we found that many of these sites are accessible as early as nc12, with a progressive acquisition of accessibility with time within this restricted time window (Figs. 4a and S4a).

To investigate whether these accessible insulator sites preferentially interacted before nc14, we performed APA analysis for nc12/nc13 (pre-ZGA), nc14 (ZGA) and 3–4 h post fertilization (hpf) (post-ZGA). As expected, preferential interactions between TAD borders first appear at nc14 and are sustained thereafter (Fig. 4b), consistent with previous analyses[20]. Thus, interactions between insulator-bound regions occupying TAD borders arise at the same time as TADs.

Next, we quantified the timing at which preferential interactions between non-border, insulator-bound regions emerged. For this, we performed APA analysis for non-borders for different developmental timings. Notably, we found that preferential interactions between non-border regions bound by insulators were already present in nc12 embryos for most Class I IBPs (Figs. 4c, d). We note, however, that further studies will be required to fully establish whether these sites are actually bound by Class I IBPs at these early stages of development.

To determine if interactions between Class I IBPs occurred at similar or reduced frequencies before nc14, we performed Hi-M imaging at nc12. The overall structure of the *dpp* locus displayed relatively minor changes between these two nuclear cycles (Fig. 4e, f), which agree with those expected from the emergence of TADs at nc14 (Figs. 4f and S4b–d). To better dissect how the proximity between insulator barcodes changed between nc12 and nc14, we calculated the proximity frequency versus cutoff distance curves for insulator and control regions (Fig. 4g). This analysis reveals that Class I IBPs co-localize with each other with similarly low frequencies in nc12 and nc14 embryos (12.83% vs 12.19% respectively). Thus, preferential interactions between Class I IBPs can be detected before the ZGA, and occur at similarly low frequencies.

To further investigate the origin of these weak interactions, we performed APA analysis from nc14 embryos treated with triptolide and alpha-amanitin, two small-molecule inhibitors of RNA Pol II activity[20]. Notably, preferential interactions between non-borders increased under these chemical perturbations (Figs. 4i and S4e). In contrast, interactions between TAD borders were relatively undisturbed (Fig. 4h). The increase in interactions between non-border insulator-bound regions is consistent with enhanced inter-TAD interactions[20] (Fig. S4f). RNA Pol II activity in these embryos is inhibited before they are transcriptionally active, thus our result indicates that preferential interactions between non-border, insulator-bound regions do not seem to require active transcription.

Finally, to shed light onto the mechanism of preferential interactions between non-border IBP sites, we performed APA analysis on embryos depleted in Zelda, a pioneering factor involved in establishing early accessibility of *cis*-regulatory elements[43]. Surprisingly, preferential interactions between non-border IBP sites were overall unaffected in Zelda-depleted embryos (Figs. 4i and S4e), suggesting that binding of class I insulators to non-border regions may not require chromatin opening by Zelda. To test this hypothesis, we first calculated the fraction of class I IBP binding sites overlapping with Zelda sites. This analysis revealed that only ~14% of the class I IBP sites corresponded to Zelda sites (Fig. 4j). Next, we calculated the accessibility of class I IBP sites at nc14 for all sites and for two subclasses: sites not bound by Zelda, and sites also bound by Zelda (Fig. 4k, l). Sites displaying both Class I IBPs and Zelda binding exhibited high accessibility, as expected. Notably, accessibility of Class I IBP sites not overlapping with Zelda represented the majority of sites and displayed significant accessibility. Overall, these results explain why preferential contacts between Class I insulators are not affected by Zelda depletion, and suggest that this class of insulators rely on other means to access chromatin during early embryogenesis.

## Discussion

In this study, we applied bioinformatic analysis to investigate the role of *Drosophila* insulator binding proteins in the folding of the zygotic genome during early embryogenesis, and combined it with novel imaging-based chromosome conformation capture approaches to quantify the absolute frequency and specificity of pairwise and multiway chromatin interactions involving insulators.

*Drosophila* insulator proteins are highly enriched at TADs borders[4,6,19,21] and contribute to the insulation of TADs[44–46]. Our bioinformatics analysis reveals that most preferential chromatin interactions genome-wide involve regions bound by class I insulators that do not involve TAD borders (>90%). This finding suggests that class I insulators are likely also involved in modulating interactions within TADs and across TAD boundaries. Members of the

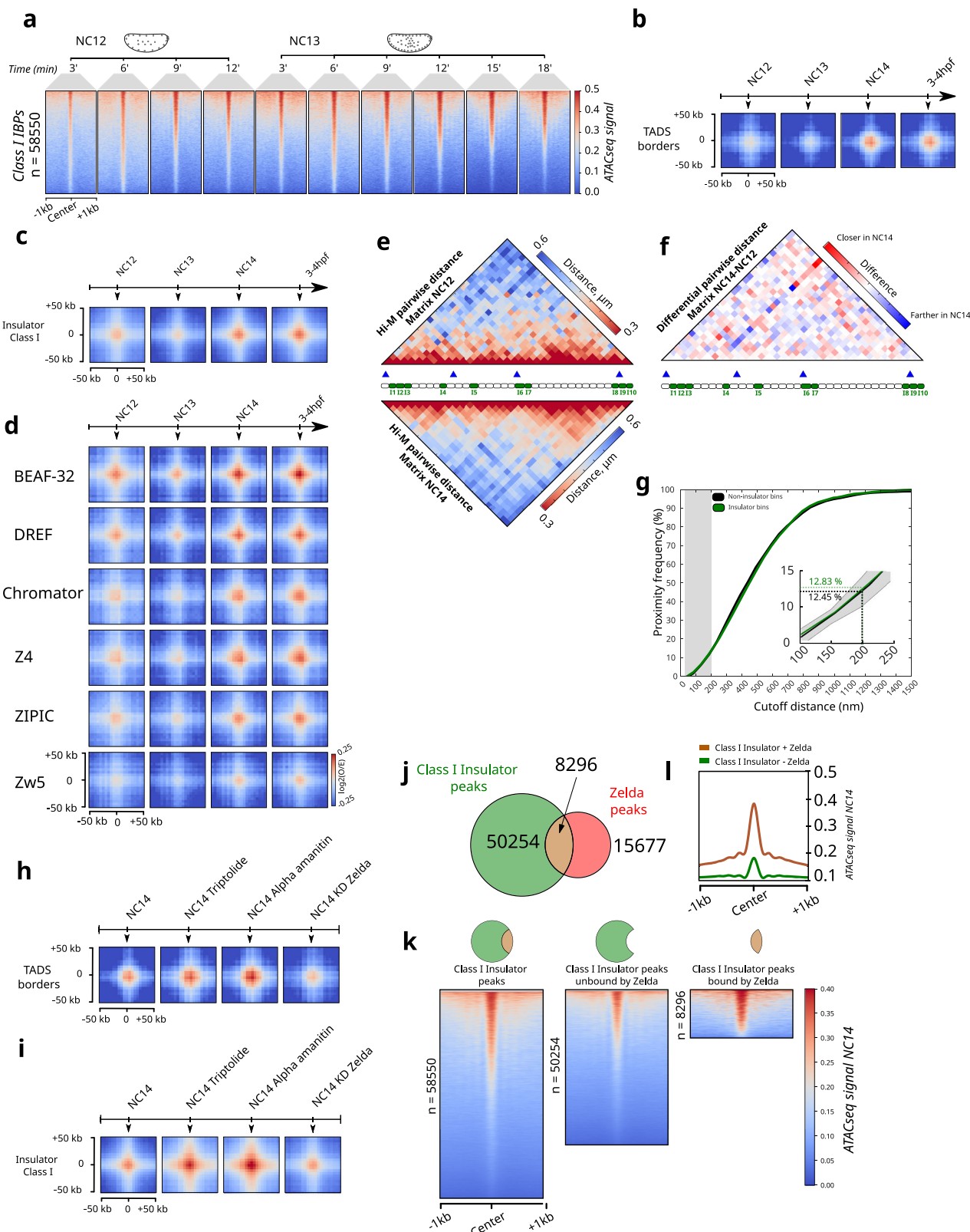

class I insulator group (e.g. BEAF-32) tend to co-localize with promoter regions[47,48] and tend to demarcate differentially-expressed genes[49], suggesting that class I insulators may play a role in modulating contacts between *cis*-regulatory modules within and between TADs. Direct promoter regulation and reduction in TAD insulation can only account for a minority (20%) of the genes downregulated upon depletion of BEAF-32[45]. Non-border chromatin

interactions by Class I IBPs appear before TADs and the onset of the zygotic genome activation, suggesting that they may contribute to defining pre-established topologies to demarcate *cis*-regulatory networks. *Drosophila* homologous chromosomes are often paired, and several factors, including insulators, play a role in this process[50], therefore contacts between insulators bound to different homologous chromosomes could also contribute to *cis*-regulation[51].

**Fig. 4 | Border–border interactions are formed at nc14, while non-border interactions are gradually formed during development. a** Series of heat maps showing the ATAC-seq signal for Class I IBPs regions across two developmental stages (nc12 and nc13), within a window of ±1 kb. **b** Aggregation Hi-C plots for TADs borders at different developmental stages (nc12, nc13, nc14, 3–4 hpf). **c** Aggregation Hi-C plots for non-border Class I IBPs group at different developmental stages (nc12, nc13, nc14, 3–4 hpf). **d** Aggregation Hi-C plots for each individual protein of the non-border Class I IBPs group at different developmental stages (nc12, nc13, nc14, 3-4 hpf). **e** Hi-M pairwise distance (PWD) matrices for nc12 constructed with 1792 traces from 4 embryos and nc14 embryos constructed with 23531 traces from 22 embryos are shown at the top and at the bottom, respectively. **f** Differential pairwise distance matrix between nc14 and nc12. Red and blue respectively represent closer and farther distances in nc14 as compared to nc12.

**g** Cumulative proximity frequency versus different cutoff distances curve for class I IBPs barcodes (green) and for 10 sets of control barcodes (black) for nc12 embryos. For the control, the solid black line represents the mean and the gray shade represents two standard deviations calculated from the variability of controls. **h** Aggregation Hi-C plots for TADs borders for different biological conditions and treatments (nc14 triptolide-treated, nc14 alpha-amanitin-treated and nc14 knockdown of Zelda). **i** Aggregation Hi-C plots for non-border Class I IBPs group for different biological conditions and treatments (nc14 triptolide-treated, nc14 alpha-amanitin-treated and nc14 knockdown of Zelda). **j** Venn diagram representing the overlap between Class I IBPs peaks and Zelda peaks. **k** Series of heat maps showing the ATAC-seq signal for the different groups of peaks shown in (**j**). **l** Metagene profiles of the ATAC-seq signal for Class I IBPs peaks bound by Zelda (orange) and not bound by Zelda (green). Source data are provided as a Source Data file.

Zelda plays a central role in rendering the zygotic genome accessible[52–55]. However, we found that interactions between Class I IBPs at non-border regions are not affected by the depletion of Zelda. This surprising result may be explained by our finding that a significant portion of class I IBPs peaks (~90%) are open at early developmental cycles (e.g. nc12) but do not co-localize with Zelda, suggesting that other unidentified pioneering factors may be required to provide access to most Class I IBPs.

Despite the genome-wide enrichment of IBPs at regions displaying 3D preferential interactions, the quantification of absolute proximity frequencies using Hi-M shows that insulator-bound regions (borders and non-borders) physically co-localize in space infrequently (~12%), and marginally more frequently than neighboring genomic regions (10.8%). This observation is consistent with low proximity frequencies between TAD borders measured in S2 cells (~10%)[37]. The low proximity frequencies between insulator-enriched regions are consistent with a recent study showing that depletion of insulators only partially weakens the strength of TAD borders[45], and with the overall absence of "focal loops" involving class I insulators in Hi-C contact maps[4,6,20,31,56]. Finally, our genome-wide analysis shows that interactions between insulator-bound regions are on average preferential, but highly variable and often weak.

The early discovery of insulator bodies led to the proposal that insulators mediate the formation of stable, rosette-like hubs involving multiple insulator-bound genomic regions[57–60]. More recently, it was shown that CP190 and Su(HW) insulator bodies formed in cultured-cells under stress conditions exhibit liquid-liquid phase separation properties[61]. This model predicts that genomically close insulators should interact in space often, nucleating interactions between multiple partners. In contrast, we observed low frequencies of pairwise proximities that rapidly decrease with the number of interacting partners (<5% for 3-way interactions and <1 % for 4-way interaction). Therefore, these results do not provide support for a widespread role of stable insulator hubs or LLPS-mediated insulator bodies in the 3D organization of the *Drosophila* genome, at least in normal physiological conditions at the *dpp* locus.

Previous studies proposed a role for *Drosophila* IBPs in mediating distant interactions[13–15,62]. Our genome-wide analysis and imaging data are inconsistent with stable interactions between class I IBPs, and suggest that these insulators may play a role at stabilizing 3D distant chromatin conformations arising from other processes, including polymer dynamics[63,64]. It is well established that binding peaks from multiple insulators often cluster together[16,49]. In this scenario, combinatorial binding of multiple insulator binding sites at single genomic locations[19,45] would provide a means to modulate the strength of the stabilization, to regulate its specificity, and to enable a locus to time-share 3D interactions with multiple genomic locations in an asynchronous manner. Consistent with this concept, analyzing binding of RNAPII and polycomb members in mouse embryonic stem cell promoter-centered chromatin interactions using network measures such as bridgeness and betweenness centrality, it was suggested that

RNAPII-bound chromatin fragments would belong to multiple communities at once, whereas polycomb bound fragments appeared to participate in multiple interactions at once[26].

Direct measurements of residence times have, unfortunately, not been reported for class I *Drosophila* insulators. However, recent studies showed that GAF and mammalian CTCF can remain bound to their cognate chromatin sites for minutes[65,66], and that CTCF loops are dynamic[67,68]. These data are consistent with a model whereby insulators help modulate the dynamics of specific interactions between distant cis-regulatory regions, but do not form stable scaffolds. These transient structures, however, may be more stable than the typical residence time of transcription factors (~10 s)[69]. In this picture, insulators could help promote transcription by stabilizing transient cis-regulatory interactions to allow for the rapid binding and unbinding of transcription factors, or rather contribute to transcriptional repression by promoting 3D conformations that prevent functional interactions. This said, the lack of clear focal peaks, the high variability in interaction strength genome-wide, and the low proximity frequencies between class I insulator-bound regions, argue for the involvement of additional molecular actors in the 3D regulation of transcription.

Finally, the methods used in this manuscript to show that *Drosophila* insulators only moderately increase the frequency of border and non-border chromatin interactions may be used to investigate insulator mechanisms in other organisms.

## Methods

### Drosophila stocks and embryo collection

The *yw* fly stocks were maintained either in a 21 °C room or in a 25 °C incubator with a natural light–dark circadian cycle. Following a pre-laying period of 16–18 h in cages with yeasted apple juice agar plates, flies were allowed to lay eggs during 1.5 h on new plates. Layed embryos were then incubated at 25 °C for an extra 2.5 h to reach the desired developmental stage. Embryos were collected and fixed as previously described[39]. Briefly, embryos were dechorionated with 2.6% freshly opened bleach for 5 min and thoroughly rinsed with water. Then, embryos were fixed in 10 mL of a 1:1 mixture of fixation buffer (4% methanol-free formaldehyde in PBS and heptane). They were then agitating for 25 min at RT. The bottom formaldehyde layer was replaced by 5 mL of methanol and embryos were vortexed for at least 30 s. Embryos that sank to the bottom of the tube, devitellinized, were rinsed three times with methanol. Embryos were then stored in methanol at −20 °C until further use.

### Hi-M libraries

Oligopaint libraries were constructed as in previous studies[25,32,39]. Briefly, each oligo had an homology region of 35–41 nt followed by a flap encoding a sequence complementary to the readout probes. We selected 138 genomic regions of interest (barcodes) in the *dpp* locus (2L:2343645..2758688 BDGP *Release 6* + ISO1 MT/*dm6*). For each barcode we used ~50 probes, covering ~3 kb. The coordinates of the targeted genomic regions are listed in Supplementary Data 2. Each

oligonucleotide in the pool (CustomArray) consisted of 5 regions: (i) a 21-mer forward primer region; (ii) two 20-mers separated by an A sequence for the barcoding; (iii) a 35/45-mer genome homology region; (iv) an extra 20-mer readout region for barcoding; and (v) a 21-mer reverse priming region. The designed oligonucleotide pools were ordered from CustomArray. The procedure to amplify a given library from the pool was previously described[39]. Briefly, the seven-step strategy consist of (i) emulsion PCR (emPCR) to extract the desired library from the pool using specific couple of primer; (ii) limited-cycle PCR from the emPCR product to determine the optimal amplification cycle; (iii) large-scale PCR with T7 promoter on the reverse primer; (iv) in-vitro transcription using T7 RNA polymerase; (v) reverse transcription; (vi) alkaline hydrolysis; and (vii) purification and concentration of the ssDNA. The sequences of the primers used for amplification of the library are listed in Supplementary Data 3.

For imaging, we used a combination of 4 barcodes to cover ~12 kb, the list of positions of the barcodes are listed in Supplementary Data 4. Each adapter consists of a 20-mer region complementary to the readout sequence that can recognize the barcode bind to a unique Alexa Fluor-647-labeled oligonucleotide (containing a disulfide linkage). Between each cycle, the fluorophore attached via a disulfide linkage can be cleavable by the mild reducing agent tris(2-carboxyethyl)phosphine (TCEP), as previously described here[39]. For fiducial, we used an adapter complementary to the reverse primer that can be bound by an unique Atto 550 labeled oligonucleotide. The sequences of the adapters and labeled barcodes purchased from Integrated DNA Technology (IDT) are listed in Supplementary Data 5.

## Hybridization of Hi-M primary library
The ssDNA library is hybridized to the DNA as previously described[39]. Briefly, embryos were rehydrated and permeabilized by sequential dilution of methanol with 0.1% Tween-20 PBS (PBT): 90%MeOH; 70% MeOH; 50%MeOH; 30%MeOH; 100%PBT (5 min each). Embryos were RNase A treated during 2 h, permeabilized 1 h with 0.5% Triton in PBS and rinsed with increased concentration of Triton/pHM buffer. pHM (pHM = 2X SSC, $NaH_2PO_4$ 0.1 M pH = 7, 0.1% Tween-20, 50% formamide (v/v)): 20%pHM; 50%pHM; 80%pHM; 100%pHM (20 min each). Then, 225 pmols of ssDNA were diluted in 25 µL of Fish Hybridization Buffer (FHB = 50% Formamide, 10% dextran sulfate, 2X SSC, Salmon Sperm DNA 0.5 mg/mL). The ssDNA and embryos were preheated at 80 °C during 15 min in separated tubes. The supernatant of the embryo's tube (pHM) is removed and the 25 µL of FHB containing the ssDNA is added. Next the mixture is transferred in a PCR-tube and deposited in the thermomixer set at 80 °C. Immediately, the thermomixer is set to decrease to 0.1 °C/min until it reaches 37 °C for an overnight incubation. The next day, the embryos were transferred to a new 1.5 mL eppendorf tube and washed two times at 37 °C during 20 min with 50% formamide, 2X SSC. Next, embryos were sequentially washed at 37 °C for 20 min with serial dilutions of formamide/PBT: 50% formamide/2× SSC; 40% formamide/2× SSC; 30% formamide/70% PBT; 20% formamide/80% PBT; 20% formamide / 80% PBT; 10% formamide/90% PBT; 100% PBT. An additional crosslink step with PFA 4% was performed and labeled embryos were washed, resuspended in PBS and stored at −20 °C for months until further use.

## Imaging system
Experiments were performed on a homemade imaging setup built on a RAMM modular microscope system (Applied Scientific Instrumentation) coupled to an improved microfluidic device, as the one described previously[39]. Software-controlled microscope components, including camera, stages, lasers, needles, pump and valves, were run using Qudi-HiM, an homemade software developed in python[70] (RRID: record ID: SCR_022114). Embryos were imaged using an ×60 Plan-Achromat water-immersion objective (numerical aperture = 1.2; Nikon) mounted on a closed-loop piezoelectric stage (Nano-F100, Mad City Labs Inc.).

The Illumination was provided by three lasers (OBIS-405 nm and Sapphire-LP-561 nm from Coherent and VFL-0-1000-642-OEM1 from MPB communications Inc.) and the images were acquired using an sCMOS camera (ORCA Flash 4.0V3, Hamamatsu, Japan). A homemade autofocus system was used to correct for axial drift in real time using a 785 nm laser (OBIS-785nm from Coherent).

## Acquisition of Hi-M datasets
Embryos were aligned on a 2% agar:PBS pad, attached to a 1:10 poly(L-lysine):water coated coverslip and mounted into a FCS2® flow chamber (Bioptechs, USA). ~20–30 embryos were selected and imaged using two regions of interest (ROI 200 × 200 µm²). Then, a mixture containing the fiducial adapter (25 nM Atto 550 imager probe, 25 nM of adapter to the reverse primer, 2× SSC, 40% v:v formamide) was injected in the chamber and let incubate for 15 min to allow complete hybridization on the primary FISH library. Embryos were washed for 10 min with a washing buffer solution (2× SSC, 40% v:v formamide) and for 5 min with 2× SSC before injecting 0.5 µg ml⁻¹ of DAPI in PBS to stain nuclei. Prior to imaging, the imaging buffer (1× PBS, 5% w:v glucose, 0.5 mg/ml of glucose oxidase and 0.05 mg/ml of catalase) was injected to reduce photobleaching of the fiducial barcode. A stack of images was acquired for DAPI and the fiducial tagged with Atto 550 (z-step size of 200 nm and a total range of 20 µm) using 405 nm and 561 nm sequential illumination. Next, the sample was sequentially hybridized as follows. A solution containing the barcode and the imager oligo was injected (25 nM Alexa-SS-647 probe, 25 nM barcode, 2× SSC, 40% v:v formamide) and incubated for 15 min. Then, the embryos were washed with 1.5 mL of washing buffer and with 1.5 mL of 2× SSC before injecting the imaging buffer. In each cycle, fiducials and readout probes were sequentially imaged with 561 nm and 647 nm excitation lasers. After imaging, the fluorescent tag of the readout probes was cleaved and discarded using 1 mL of chemical bleaching buffer (2× SCC, 50 mM TCEP hydrochloride). Finally, samples were washed with 1 mL of 2× SSC for 5 min before a new hybridization cycle started. Further details can be found on our previously published protocol[39].

## Image processing
DCIMG files were converted to TIFF using proprietary software from Hamamatsu. TIFF images were then deconvolved using Huygens Professional 21.04 (Scientific Volume Imaging, https://svi.nl). The analysis was performed using our pyHiM analysis pipeline (https://pyhim.readthedocs.io/en/latest/). Briefly, images were first z-projected using either sum (DAPI channel) or maximum intensity projections (barcodes, fiducials). Fiducial images from each hybridization cycle were used to register barcode images using global and local registration methods. Next, barcode images were segmented in 3D using stardist[71] and the positions of the centers of barcodes were detected with sub-pixel resolution using Big-FISH (https://github.com/fish-quant/big-fish)[72]. DAPI images were segmented in 3D using stardist. Barcodes were then attributed to each single nucleus mask by using their XY coordinates. Finally, pairwise distance matrices were calculated for each single nucleus. From the list of pairwise distance maps, we calculated the proximity frequencies as the number of nuclei in which pairwise distances were within 200 nm normalized by the number of nuclei containing both barcodes. Hi-M maps of nc14 embryos were generated from a total of 23531 traces from 22 embryos from 2 separate experiments. The maps for nc12 embryos are constructed from 1792 traces from 4 embryos from 2 separate experiments.

## Insulation score derived Hi-M dataset
Insulation scores derived from the Hi-M dataset were computed by moving an n-by-n square window along the diagonal of the median pairwise distance and summing the distances within this square.

Domainogram were calculated by smoothing a matrix obtained by computing the IS with an increased window size (from 1-by-1 to 6-by-6) over the Hi-M matrix.

## Multiway proximity frequency analysis

The proportion of multiway contacts is calculated from single nucleus proximity frequency nc14 matrices[73]. Briefly, we counted the number of multiway contacts where the selected anchor barcode was interacting with other partners within a 200 nm radius. These values were normalized by the number of pairwise interactions for each anchor. The expected proximity frequency is derived by considering all events as independent. For this, we computed the mean of the product of all possible barcode combinations for various numbers of interacting partners.

## Chip-Seq data processing

Insulator proteins ChIP-Seq fastq files were downloaded from Gene Expression Omnibus (GEO) with GSE62904 and GSE54337 primary accession numbers. The quality of the reads was estimated with FastQC[74] (0.11.7). Sequencing reads were aligned to the reference *Drosophila melanogaster* genome assembly (dm6) using Burrows–Wheeler Aligner[75] (0.7.17-r1188) with default parameters. Finally, peak calling was performed using MACS2[76] (2.2.7.1) with default parameters. BEAF-32, ZELDA, Zw5, PolIISer5 and Pc / Ph raw data were downloaded from GEO under series accession code GSE62904, GSE30757, GSE76997, GSE62925, GSE60428 respectively, and processed as previously described. ChIP-on-Chip insulator proteins data from GSE26905 GEO series have been downloaded as bed files and peak coordinates have been converted from dm3 to dm6 by using FlyBase's sequence coordinates converter (FB2021_04, released August 17, 2021). The accession numbers of the data used in this study are listed in Supplementary Data 6.

## ATAC-Seq data processing

ATAC-seq data were downloaded from GSE83851[42]. *Wig* files were converted to *BigWig* using *wigToBigWig* from UCSC. Heat maps of ATAC-seq profiles were then plotted over ±1 kb window centered on Class I IBPs sites using *computeMatrix* followed by *plotProfiles* from *deepTools*[77]. Average ATAC-seq profiles derived from the heat maps for individual IBPs were constructed using a custom Matlab script. Venn diagrams between Class I IBPs and Zelda peaks were generated from bed files using *Intervene*[78] and plotted using a custom python script. Different bed files coming from each group were generated using *intersect* and *subtractBed* from bedtools v.2.3.

## Boundary calling

We used the previously annotated list of TAD boundaries from Hug et al.[20] Briefly, boundaries were called using the insulation score metric defined by Crane et al.[79] using a 5 kb balanced contact matrix with a window size of 8 bins.

## Chromatin assortativity

In order to build networks needed for Chromatin Assortativity, Hi-C contact matrices were used with a 5 kb resolution. Chromosight[29] (1.3.3) was used with different sets of parameters to create different networks. Network 1 was built with the following parameter set: --pearson 0.3, --min-dist 20 kb, --max-dist 2 Mb, --min-sep 5 kb, --max_perc_0 10. Network 2 was built using: --pearson 0.3, --min-dist 10 kb, --max-dist 200 Mb, --min-sep 5 kb, --max_perc_0 50. Network 3 was built using: --pearson 0.2, --min-dist 10 kb, --max-dist 200 Mb, --min-sep 5 kb, --max_perc_0 50. In all cases, Chromosight was used with the "--norm" parameter set to "auto" to instruct Chromosight to use matrices normalized using the Knight–Ruiz balancing algorithm[80]. As a pre-processing step, Chromosight normalizes Hi-C matrices by genomic distance using observed/expected values. Significant

chromatin loops are called on these normalized matrices. A specific genome scale chromatin network was built where nodes are genomic fragments and edges are significant interactions between two fragments. Then, chromatin networks were loaded on R and ChIP-seq peaks were used as features assigned to nodes using the ChAseR R package[81] (0.0.0.9) to calculate chromatin assortativity. For each feature, 1000 randomized networks preserving genomic distances and corresponding chromatin assortativity values were computed.

The chromatin assortativity Z-Scores calculation is given by the following formula:

$$z = \frac{X - \mu}{\sigma} \tag{1}$$

where $X$ is the feature chromatin assortativity value, $\mu$ is the randomizations ChAs average and $\sigma$ is the standard deviation from the randomization distribution.

The calculation of the Z-Score allows estimating the significance of the assortativity values with respect to the assortativity expected based purely on correlation of feature values along the linear genome. For example, domains of a feature that span multiple bins in the Hi-C matrix are more likely to produce high ChAs values, but this does not imply the importance of the 3D contacts.

Cytoscape (3.8.0) was used for chromatin network visualization. Methods for Cross-ChAs and AND-ChAs are provided elsewhere[27].

## Log2(O/E) and Hi-C aggregate plot analysis

ChIP-seq peaks were used to extract a list of regions bound by a set of putative factors. Next, for each autosomal arm, we computed the log2(Observed/Expected) Hi-C contact value by calculating the average contact frequency for all the combinations of regions separated by a certain genomic distance on a 5-kb-Hi-C dataset[20] (E-MTAB-4918). We then selected the interaction between the regions bound by the set of factors on the distance-normalized Hi-C dataset. The distribution of log2(O/E) between all pairwise combinations of regions bound by the investigated factors was then displayed in a violin plot or divided into equal-size groups depending either on protein occupancy or insulation score and displayed in Hi-C aggregate plots.

Hi-C aggregate plots were performed using a homemade analysis pipeline developed in MATLAB Release R2019b (The MathWorks, Inc., Natick, USA). The distance-normalized sub-matrices over a window of 100 kb surrounding the intersection between two anchored peaks were extracted. Finally, the aggregate plots were then created by averaging all of the sub-matrices together. For the bootstrapping method (Fig. S1o), we performed a series of iterations by randomly selecting $N$ BEAF-32 anchors from the full list of anchors ($N$ values were chosen as 2, 5, 10, 25, 50, 100, 250, and 300). We then calculated the mean log2(O/E) for this set of $N$ anchors and repeated this process 10,000 times for each value of $N$.

## Reporting summary

Further information on research design is available in the Nature Portfolio Reporting Summary linked to this article.

## Data availability

The single nucleus pairwise distance matrices as well as XYZ coordinates of chromatin traces generated in this study have been deposited at our Open Science Framework project (https://osf.io/aqtxj/) with https://doi.org/10.17605/OSF.IO/AQTXJ. The list of previously published datasets used in this study is provided in Supplementary Data 6. Source data are provided with this paper.

## Code availability

The code used for aggregation plot analysis and for post-processing Hi-M matrices are accessible at https://github.com/NollmannLab/

messina_2022. For a permanent link, see https://doi.org/10.17605/OSF.IO/AQTXJ. Hi-M data were acquired using qudi-HiM[70]. The current version of qudi-HiM is found at https://github.com/NollmannLab/qudi-HiM, and an archived version at https://zenodo.org/record/6379944 (https://doi.org/10.5281/zenodo.6379944). Hi-M data were analyzed using pyHiM release 0.6, available at https://github.com/marcnol/pyHiM.

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

## Acknowledgements

This project was funded by the European Union's Horizon 2020 Research and Innovation Program (Grant ID 724429) (M.N.). We acknowledge the Bettencourt–Schueller Foundation for their prize 'Coup d'élan pour la recherche Française', and the Drosophila facility (BioCampus Montpellier, CNRS, INSERM, Univ Montpellier, Montpellier, France). The CBS is a member of the France-BioImaging, a national infrastructure supported by the French National Research Agency (ANR-10-INBS-04-01). O.M. was supported by an FRM and Ligue Contre la Cancer PhD fellowships. F.R. and V.P. were supported by Fondation Toulouse Cancer Santé and Pierre Fabre Research Institute as part of the Chair of Bioinformatics in Oncology of the CRCT.

## Author contributions

O.M., V.P. and M.N. conceived the study and the design. O.M. acquired the data. O.M., F.R., and J.G. analyzed the data. O.M., J.-B.F., and F.R. wrote the software. J-B.F. built the microscope. O.M., F.R., V.P. and M.N. interpreted the data. M.N., O.M., and V.P. wrote the manuscript. M.N. and V.P. supervised the study and acquired funds.

## Competing interests

The authors declare no competing interests.
