## [Peer Review File · Nature Communications]

3D chromatin interactions involving *Drosophila* insulators are infrequent but preferential and arise before TADs and transcriptionREVIEWER COMMENTS

Reviewer #1 (Remarks to the Author):

The authors of this manuscript have set out to examine the role of insulators in shaping the 3D structures of the *Drosophila* genome. Specifically, they aim to uncover the impact of insulators on the formation of TADs and the formation of interactions within and across TAD boundaries. To accomplish this, they employ a combination of Chromatin Assortativity Analysis (ChAs) and Aggregation Peak Analysis (APA) on Hi-C data, along with multiplex imaging experiments to observe the frequency of insulator interactions and multi-way contacts in individual cells.

The main takeaways from the study are: 1) genomic loci bound by insulator proteins tend to cluster together in 3D space, 2) insulator-insulator contacts aren't restricted to TAD borders and can occur within or across TADs, and 3) in single cells, insulator-insulator interactions are rare and mostly occur as pairs rather than in a rosetta-like manner.

In my opinion, the methodology used in the study is reasonable and the conclusions drawn are valuable. Although the ChAs and APA analysis of the Hi-C data is a nice touch, the findings regarding preferential interactions among insulators are not entirely new and have been reported in previous studies. However, the multiplex imaging experiment does provide novel insights that cannot be obtained from the Hi-C data alone, such as the absolute frequency of pairwise and multi-way contacts. I think that further analysis and discussion of the imaging data would provide additional useful information. Please see my detailed comments and questions below.

- It's important to note that a significant portion of the analysis in this study hinges on the identification of TAD borders, so it would be beneficial to provide a clear explanation of the computational methods used to determine TADs and their boundaries in the Methods section. Additionally, it would be helpful to discuss the robustness of the results and conclusions with regard to the parameters employed in determining TADs and their borders.
- In line with the previous comments, it would be helpful to include more information about Chromatin Assortativity. The study mentions that ChromoSight was used to identify significant interactions using default parameters. If the authors believe that the default parameters are sufficient for the purposes of this study, it would be good to provide some justification for this choice.
- Following up on the previous comment, I was wondering if the Hi-C contact map was normalized by the genomic distance prior to being used in the ChAs analysis. Perhaps this is handled by ChromoSight, but it would be helpful if the authors could provide clarification on this matter.
- Line 132-134, it states that "Positive correlation between indicates that ... chromatin factor ... most often increase the strength of ...". I believe the correlation does not necessarily imply causality. It is possible that high ChAs and high $\log_2(O/E)$ values could stem from the same source rather than one causing the other.
- At the end of the section "Genomic regions displaying preferential interactions are predominantly bound by chromatin insulators", the authors discussed the ChAs Z-score for GAF and claimed that its low ChAs Z-score is an "artifact" of the ChAs analysis. However, even the APA shows negative $\log_2(O/E)$ values for GAF (Fig. 1c), which aligns with its ChAs Z-score. These results appear to contradict previous findings cited in references 29 and 31. Could the authors provide an explanation for this discrepancy?
- At the end of the section "Genomic regions displaying preferential interactions are predominantly bound by chromatin insulators", it is stated that "ChAs do not shed light on whether and how binding of insulator may modulate interaction frequencies". However, I believe that the ChAs values do indeed reflect the interaction frequency of the insulator. This is because significant interactions are used to construct the network, so it seems logical to me that high ChAs should be linked to high $\log_2(O/E)$ values. Figure 1c demonstrates that these two quantities are strongly correlated and effectively convey the same information. Could the authors provide their thoughts on this?

- Line 174-176, it states that preferential interactions involving borders are a minority of all preferential interactions, 1% for border/border etc. It would be intriguing to see the impact of normalizing the numbers of borders and non-borders on these percentages. Given that the number of borders is likely much smaller than that of non-borders, the percentage of border/border interactions would be greatly impacted. Normalizing with respect to their numbers could shed light on whether the likelihood of border/border interaction is higher or lower than expected.

- Line 205, "ensemble pairwise distance maps were built by kernel density estimation of the full pairwise distance". Since the ensemble pairwise distance maps are simply the average distances map, I'm curious as to why the authors opted for using kernel density estimation instead of simply calculating the mean values of the distances. The ensemble pairwise distance maps appear to be an average of the distances, so what additional benefits does the kernel density estimation provide?

- When I compare the average distance map in Fig. 3c to the Hi-C map in Fig. 3b, it seems like the average distance map has much more noise and the TAD structures are not as clear as in the Hi-C map. Have the authors calculated the standard error for the average distances? How many samples were used in the calculation and what level of confidence can be placed on the interactions marked by the yellow and green arrows in Fig. 3c?

- One of the findings in the study is the absence of rosette-like structures among insulators, due to the low probability of having multiple contacts between insulators. I concur with this conclusion as the imaging data in this study, as well as previous studies, show that at any given cell, the probability of multiple loci being close together is low, which is understandable as the probability of having multi-way contacts decreases when pairwise interactions are not strongly dependent. But, I think a different angle could be considered as well. Is the probability of multi-way contact higher or lower than expected? This idea is similar to the Observed/Expected analysis commonly performed in Hi-C data. By computing the multi-way (at least three-way) interactions from imaging data, a more informative result could be obtained, and potentially highlight "significant" multi-way contacts and interactions.

- Regarding the statement on line 264-266 about the overall structures of the dpp locus considerably changing, it would be helpful to see a different comparison or plot for better visualization.

- Line 269-271 states that insulator colocalization is present as early as nc12, but when I look at Fig. 4b, it seems to suggest otherwise. The proximity probability of insulators and non-insulators is almost the same (12.5% vs. 10.9%). This raises questions about whether these interactions are actually stabilized at nc12 or if they just happen due to polymer fluctuations. Can the authors provide clarification on this issue?

- On line 286-288, it states that "the increase in interactions between non-border insulator-bound regions is consistent with enhanced inter-TAD interaction". To properly support this statement, it would be necessary to first separate the interactions between non-border insulators into inter-TAD and intra-TAD, and then focus only on the inter-TAD interactions among non-border insulators.

Minor comments:

- The resolution of figures in the SI is too low. For some of the figures, the texts and numbers are unreadable.

- The x-axis label for Fig. 1 C should be "mean of $\log_2(O/E)$ " or something like that

- XYZ coordinates data from the imaging experiment should be provided.

- The details of how the insulation score (IS) is calculated should be provided in the methods section or SI

- The distribution of $\log_2(O/E)$ shown in Fig. S1c seems to be multi-modal. Does this indicate that there may be sub-classes of a given IBP?

Reviewer #2 (Remarks to the Author):

This manuscript by Messina et al. is addressing an important question about the roles of insulator proteins in *Drosophila*. The combination of computational analysis of chromatin assortativity scores with *in vivo* Hi-M has the potential to provide significant insights into this topic. However, the chromatin assortativity analysis lacks sufficient quality controls. In addition, there are several points where the authors overstate their findings or where the phrasing is unclear. Overall, the paper requires additional analyses to clarify and confirm the robustness of their findings.

Major points

1. The Chromatin Assortativity analysis relies on the input network being robust and reliable. Here, the input network is based on loop calls by Chromosight, but no analysis of the quality or robustness of these loop calls is performed. This is particularly important since *Drosophila* loops have different characteristics from mammalian loops (e.g., they are not typically found at domain corners), which affect the ability of loop callers designed for mammalian data to detect them. Chromosight does not appear to have been tested on *Drosophila* data in the original publication. Therefore, to support the use of this network for downstream analysis, it is essential that the authors include an analysis of the quality and robustness of the loop calls used for the network, including typical loop sizes and how well their loop calls overlap with those from other studies (e.g. Batut et al. 2022, PMID: 35113722), as well as visualisations of example loci with loops.
2. The insulator ChIP-seq datasets used are from cell lines, while the Hi-C data is from early embryos. The limitations of the data should be discussed. E.g. while interactions between regions bound by these insulators appear before TADs, it's not clear whether these IBPs are actually bound at nc12. Additional datasets from embryos and/or analysis of chromatin accessibility data at nc12 would support the idea that the IBPs actually bind to these loci at this timepoint.
3. None of the factors tested have a negative ChAs score – are any chromatin proteins expected to be disassortative? How was the threshold of 2 chosen to select those for further analysis?
4. The aggregate peak analysis has been carried out using all pairs of sites – is there a difference if using only pairs within a certain distance? It would be interesting to see if some factors have preferential interactions only at short distances, which are not seen when including long-range pairs.
5. To better link the two halves of the paper, it would be helpful to show the loops, if any, in the *dpp* locus, that were called with Chromosight and used for the network analysis.
6. The authors conclude that multi-way interactions are rare. Would it be possible to calculate how often triplet/quadruplet/etc interactions would be expected by chance, given the observed pairwise interaction frequencies? It would be informative to know if multiway interactions occur more, less, or equally as often as expected by chance.

Minor points

1. The introduction states that *Drosophila* insulators “do not seem to rely on loop extrusion”, yet this doesn't follow from the previous paragraphs – loop extrusion itself can occur without CTCF-mediated blocking. The authors should rephrase or clarify this point.
2. The explanation of how low assortativity can arise (lines 116-118) is unclear – a schematic might help. Because of this I find that the justification for low assortativity of GAF is not well-explained. The authors should clarify this part of the text. As mentioned above, it would also be very informative to check the overlap of their loop calls with the GAF-bound tethers identified by Batut et al.
3. Line 132-134: the current phrasing implies that chromatin factors with positive ChAs scores cause increased interaction strength, however the data presented cannot show causation. Please rephrase this sentence.
4. Fig 2a: It's unclear to me whether this a schematic or shows the actual network. If this shows the actual network, it doesn't seem to be consistent with the very low % of border-border interactions in Fig 2e. The authors should clarify this.
5. Lines 186-187: the strong interaction seen in Fig 2f does not imply anything about the binding of IBPs, but rather is simply consistent with the aggregate analysis of called loops. This sentence should be rephrased to clarify what the authors want to show here, or the analysis should be removed.

6. I think line 231 has a typo - non-insulator barcodes colocalise at a slightly lower frequency than insulator barcodes.
7. I find it hard to identify/appreciate the key differences in Fig 4a. It would be helpful to show the reference plots for nc14 and nc12 as well.
8. Line 269 – what is meant by “specificity” here? How is this measured?
9. Line 318 – the wording “class I insulator family” might inadvertently imply an evolutionary relationship between these proteins.

Reviewer #3 (Remarks to the Author):

In the manuscript “3D chromatin interactions involving *Drosophila* insulators are infrequent but preferential and arise before TADs and transcription” Messina and colleagues combined bioinformatics analysis and multiplexed chromatin imaging to unveil the contribution of insulators in shaping the 3D genome organization in *Drosophila*, that intriguingly differ from mammals one in varied aspect. Characterizing these differences better is an interesting and fundamental question in the field that, interestingly, is also accumulating provocative and contradicting findings. Generally, the presented data are, for the most part, convincing, although in some sections, the manuscript lacks the necessary clarity to convince a reader that the results are robust and relevant. Additionally, I recommend toning down the title and making it less generic, as the work presented supported by HiM experiment focuses on a small genomic region spanning only 3 TADs. To this end, I have some concerns that need to be improved before publication.

Specific critiques need to be addressed and discussed:

- One of my concerns is related to the first section of the results, where the authors dissect the preferential interactions of a set of 15 possible insulator proteins using Chromatin Assortativity analysis (ChAs). The analysis, by design, creates a chromatin interaction network focusing only on highly frequent interactions present in the population ensemble experiment (HiC data). This network construction can compromise the results and conclusions as it does not consider the multiple low-frequency and low-affinity interactions that have a role in TAD-like structure generation (as the authors identify in previous work (Cattoni Nat Commun 2017)). Specifically, I found interesting the identification of class I and II insulators; however, it would be nice to see if this classification and the Assortativity of the network hold when also the low-frequency interactions are accounted for.
- Another aspect that has not been investigated thoroughly in this analysis but that I believe needs to be addressed to truly dissect the insulators’ role in the folding of the zygotic genome during early embryogenesis genome-wide is considering the combination of these factors. Indeed, it has been shown, for example, that the depletion of the insulator BEAF-32 does not abolish boundaries (Ramirez et al., Nat. Commun. 2018 PMID: 29335486), possibly suggesting that potentially a combination of factors is required for boundary maintenance. The author tried to address this aspect by HiM co-localization analyses (see below) but only on a small locus of ~415kb located in chr2L:2343645-2758688 and not fully genome-wide. How does the Chromatin Assortativity analysis change if the combination of class I and/or class II insulator are considered?
- In the “Insulator binding increases the strength of preferential long-range chromatin contacts” section, the author refers to long-range interaction; however, the aggregation peak analysis is reported for only 100kb regions around the anchor point, which is more related to short-range interaction rather than long-range (generally occurring at the Megabase scale). This is valid throughout the presented manuscript; as such, I recommend that the author rephrase the relative sections to enhance the general clarity of the findings.
- As the author points out in the manuscript, HiC is an ensemble-based method that cannot inform on single-cell patterns of interactions, an aspect that is possible to investigate with chromatin tracing methods such as HiM. Indeed, to account for this limitation, the author specifically imaged a small locus of ~415kb located in chr2L:2343645-2758688 subdividing the region into 34 equally spaced barcodes. 3 TADs characterize the locus in the population cell experiment (unfortunately,

the color-bar scale of Figure 3 B does not show it fully, see minor points section). Unfortunately, it is unclear if the border-containing-barcode regions have been defined by Insulation Score analysis of the pairwise matrix or proximity frequency matrix derived by HiM. If not, how do the insulation profile of the HiM-derived matrices and the HiC compare? The author reported the efficiency of detection as a percentage, but it would be nice also to have explicit quantification of the number of nuclei/ traces analyzed. To this end, it is known (also by a previous work of the author) that the nuclei in nc14 embryos are characterized by homologs showing pervasive pairing and that varied elements, including insulator elements, play a role in pairing (e.g., Rowley & Corces Nat. Rev. Genet. 2018). The author should elaborate on this aspect that delineates one of the main differences between 3D genome organization in *Drosophila* and mammals.

- One of the main aspects of HiM is that it quantifies the real co-localized at the single-cell level. The author chose ten regions as control (marked in yellow) to probe if the insulator-bound regions are spatially co-localized. How does the cumulative curve change if another set of ten regions is considered as a control? Why did the author select only those to use as a control? I believe the analysis could gain strength if different sets of control regions are considered. This is needed to confirm the robustness of the minimal differences observed in the comparison between nc12 and nc14 embryos Hi-M maps when TADs emerge. That is also the main message of the manuscript title, which I recommend reconsidering since the author investigates only a specific locus.

Additional comments:

Generally, I recommend editing part of the text and checking for general consistency.

- P4 line 86. "a chromatin interaction network is built from a single genome-wide contact map", I found the use of 'single genome-wide contact map' misleading as it refers to an ensemble experiment.

- P11 lines 394-395, the sentence is missing a verb.

"Oligopaint libraries from a public database (<http://genetics.med.harvard.edu/oligopaints>), consisting of unique 35/45-mer sequences with genome homology."

- Figure 1C, green labels are unclear. Z4 appears to be of different green and class than BEAF32, while in Figure 1E are listed as the same class. From the legend's explanation, L(3)MBT and Pita should be class II, but they are color-coded differently. Please clarify.

- Figure 2H and Supplementary Figure 2C, in figure legend 2H, the "h" is capital letter and not in agreement with the rest of the text. Aside from this, the plots are not clear enough, and the annotation is not visible. The authors could elaborate better on these plots and clarify the confusion between insulation strength (based on the insulation score) and insulator binding strength (how the latter is defined?).

- Figure 3 legend is poorly described: in panel B, *Drosophila melanogaster* needs to be in italics, and in panel C, the explanation of what the arrows are is missing. Additionally, in Figure 3B the color-coded is misleading, and it is difficult to see the TADs border. Please adjust it.

- Figure 4E should be presented more explicitly and with appropriate statistics.

Point-by-point answer to reviewer's comments for:

3D chromatin interactions involving *Drosophila* insulators are infrequent but preferential and arise before TADs and transcription

Olivier Messina¹, Flavien Raynal², Julian Gurgo¹, Jean-Bernard Fiche¹, Vera Pancaldi^{2,3*}, Marcelo Nollmann^{1,*}

Affiliations

¹ *Centre de Biologie Structurale, Univ Montpellier, CNRS UMR 5048, INSERM U1054, 34090 Montpellier, France.*

² *Université de Toulouse, Inserm, CNRS, Université Toulouse III-Paul Sabatier, Centre de Recherches en Cancérologie de Toulouse, 31037 Toulouse, France.*

³ *Barcelona Supercomputing Center, Barcelona, Spain.*

* Corresponding authors: vera.pancaldi@inserm.fr, marcelo.nollmann@cbs.cnrs.fr

Color convention:

- *Reviewers comments are shown in italics*
- **Our answers are provided in green.**
- **Changes to the revised manuscript are shown in blue.**

Reviewer #1 (Remarks to the Author):

The authors of this manuscript have set out to examine the role of insulators in shaping the 3D structures of the Drosophila genome. Specifically, they aim to uncover the impact of insulators on the formation of TADs and the formation of interactions within and across TAD boundaries. To accomplish this, they employ a combination of Chromatin Assortativity Analysis (ChAs) and Aggregation Peak Analysis (APA) on Hi-C data, along with multiplex imaging experiments to observe the frequency of insulator interactions and multi-way contacts in individual cells.

The main takeaways from the study are: 1) genomic loci bound by insulator proteins tend to cluster together in 3D space, 2) insulator-insulator contacts aren't restricted to TAD borders and can occur within or across TADs, and 3) in single cells, insulator-insulator interactions are rare and mostly occur as pairs rather than in a rosetta-like manner.

In my opinion, the methodology used in the study is reasonable and the conclusions drawn are valuable. Although the ChAs and APA analysis of the Hi-C data is a nice touch, the findings regarding preferential interactions among insulators are not entirely new and have been reported in previous studies. However, the multiplex imaging experiment does provide novel insights that cannot be obtained from the Hi-C data alone, such as the absolute frequency of pairwise and multi-way contacts. I think that further analysis and discussion of the imaging data would provide additional useful information. Please see my detailed comments and questions below.

We thank the referee for their careful examination of the manuscript. We have addressed these points by implementing changes to the manuscript and figures, which are outlined below.

1.1 - *It's important to note that a significant portion of the analysis in this study hinges on the identification of TAD borders, so it would be beneficial to provide a clear explanation of the computational methods used to determine TADs and their boundaries in the Methods section. Additionally, it would be helpful to discuss the robustness of the results and conclusions with regard to the parameters employed in determining TADs and their borders.*

We apologize for the lack of clarity in the original submission.

We used a previous TAD boundary call from Hug *et al.*, 2017. The list of boundaries called is available at: <https://github.com/vaquerizaslab/Hug-et-al-Cell-2017-Supp-Site>. As explained in that publication, TAD boundaries were called using the insulation score metric defined by (Crane *et al.*, 2015) using a 5kb balanced contact matrix with a window size of 8 bins.

To clarify this, we added this information to the methods section as follows :

“Boundary calling

We used the previously annotated list of TAD boundaries from Hug *et al.* (Hug *et al.*, 2017). Briefly, boundaries were called using the insulation score metric defined by (Crane *et al.*, 2015) using a 5kb balanced contact matrix with a window size of 8 bins.”

We agree with the reviewer that it is crucial to assess the reliability of our results with respect to the parameters used to determine TAD boundaries. To address this concern, we conducted an additional TAD call using the FAN-C tool (Kruse *et al.* 2020) on the Hi-C data from Hug *et al.* 2017. We then performed a side-by-side comparison with the TAD boundary annotation from Hug *et al.* (2017).

Borders annotations	Quantification
Calling from Hug et al. (2017) (used in this ms)	Original Fig. 2e
Calling using FAN-C	

FAN-C called a total of 1929 borders, while Hug *et al.* (2017) had reported 672. In spite of this, the proportion of loops in each category (Border / Border, Border / Non-Border and Non-Border / Non-border) are similar between the two alternative TAD calling methods. Thus, our main conclusions are robust and do not depend strongly on the method used to call TADs.

1.2 - *In line with the previous comments, it would be helpful to include more information about Chromatin Assortativity. The study mentions that ChromoSight was used to identify significant interactions using default parameters. If the authors believe that the default parameters are sufficient for the purposes of this study, it would be good to provide some justification for this choice.*

To address this concern, we explained clearly the parameters used for running ChromoSight (see revised Methods below), and performed complementary analysis using other parameter sets to test the robustness of our choice.

Chromosight was set to detect loop patterns and automatically generate a configuration file with appropriate detection parameters given the data resolution and sequencing depth. In our case, the genomic distance range for loop detection was set between 20kb (4 bins) and 2Mb (400,000 bins) and the Pearson correlation threshold used to define significant interactions was set to 0.3 (see figure below). We used Hi-C matrices normalized according to the Knight-Ruiz (KR) balancing method, as in Hug *et al.* (2017). The balanced weights were automatically detected by Chromosight and reused for loop pattern detection. This is now described in the revised Methods, section “Chromatin assortativity”.

To validate the robustness of our analysis, we performed a complementary analysis with two other sets of Chromsight parameters for loop calling:

	Set 1 (Network 1, used in the ms)	Set 2 (Network 2)	Set 3 (Network 3)
Parameters used	--pearson 0.3 --min-dist 20kb --max-dist 2Mb --min-sep 5kb --max_perc_0 10	--pearson 0.3 --min-dist 10kb --max-dist 200Mb --min-sep 5kb --max_perc_0 50	--pearson 0.2 --min-dist 10kb --max-dist 200Mb --min-sep 5kb --max_perc_0 50
Number of called loops	2153	3206 (+ 48.91 % more than Network1)	17567 (715.93 % more than Network1)

Next, we calculated the ChAs Z-scores for the different insulator factors for Networks 2 and 3 (see Table below). The Z-scores for Networks 2 and 3 were highly correlated with those for the reference network (Network 1), even when these networks were considerably larger than Network 1, and contained loops with lower interaction frequencies. Thus, our results are robust with respect to the choice in Chromsight parameters.

After this comprehensive comparison between the default set of parameters and two alternative choices, we conclude that the use of default parameters in the manuscript is adequate and justified. We added the panels above (new Fig. S1f) and revised the text as follows to include these complementary analyses :

“To validate the robustness of these analyses, we performed similar analysis for different sets of Chromosight parameters (see *chromatin assortativity* Methods) generating larger networks that include lower-frequency interactions. We found that the insulator factors with the highest ChAs Z-scores were the same independently of the size of the network (Fig. S1f).”

We also provided information on the parameters used to construct the 3 different networks in the Methods section:

“In order to build networks needed for Chromatin Assortativity, Hi-C contact matrices were used with a 5 kb resolution. Chromosight²⁸ (1.3.3) was used with different sets of parameters to create different networks. Network 1 was built with the following parameter set: --pearson 0.3, --min-dist 20kb, --max-dist 2Mb, --min-sep 5kb, --max_perc_0 10. Network 2 was built using: --pearson 0.3, --min-dist 10kb, --max-dist 200Mb, --min-sep 5kb, --max_perc_0 50. Network 3 was built using: --pearson 0.2, --min-dist 10kb, --max-dist 200Mb, --min-sep 5kb, --max_perc_0 50. ...”

1.3 - *Following up on the previous comment, I was wondering if the Hi-C contact map was normalized by the genomic distance prior to being used in the ChAs analysis. Perhaps this is handled by Chromosight, but it would be helpful if the authors could provide clarification on this matter.*

As a pre-processing step, Chromosight normalizes the matrix by genomic distance by computing the observed/expected matrix. Loops are then called on these normalized matrices. This procedure is explained in the original chromosight paper (Matthey-Dorey et al, 2020 - Nature communications) : “A detrending procedure, to remove distance-dependent contact decay due to polymeric behavior, is then applied, which consists in dividing each pixel by its expected value under the polymer behavior”.

We have added this information to the Methods section as follows:

“Chromatin Assortativity

... In all cases, Chromosight was used with the “--norm” parameter set to “auto” to instruct Chromosight to use matrices normalized using the Knight-Ruiz matrix balancing algorithm. As a pre-processing step, Chromosight normalizes HiC matrices by genomic distance using observed/expected values. Significant chromatin loops are called on these normalized matrices.”

1.4 - *Line 132-134, it states that “Positive correlation between indicates that ... chromatin factor ... most often increase the strength of ...”. I believe the correlation does not necessarily imply causality. It is possible that high ChAs and high log₂(O/E) values could stem from the same source rather than one causing the other.*

We apologize for the lack of clarity in our original submission. We agree that the high correlation between ChAs Z-scores and log₂(O/E) values does not imply causality. Both methods detect preferential interactions, and this may partially explain why an insulator protein with high log₂(O/E) value will often also display a high global ChAs score. However, we stress that the two methods give different and complementary results about insulator interactions (see also **1.6** below). To avoid ambiguity, we revised the text as follows:

“Notably, the positive correlation between ChAs and log₂(O/E) (**Fig. 1c**) indicates that factors displaying high assortativities are bound to chromatin regions that exhibit the most preferential interactions.”

1.5 - At the end of the section “Genomic regions displaying preferential interactions are predominantly bound by chromatin insulators”, the authors discussed the ChAs Z-score for GAF and claimed that its low ChAs Z-score is an “artifact” of the ChAs analysis. However, even the APA shows negative $\log_2(O/E)$ values for GAF (Fig. 1c), which aligns with its ChAs Z-score. These results appear to contradict previous findings cited in references 29 and 31. Could the authors provide an explanation for this discrepancy?

We agree with the reviewer that both negative $\log_2(O/E)$ values and low ChAs Z scores seem to contradict the previous findings (Ogiyama et al., 2018 and Loubiere et al., 2020) showing that GAF is important for the formation of repressive loops. This apparent discrepancy can be explained by the fact that repressive loops involving GAF appear after the midblastula transition (Ogiyama et al., 2018), whereas our analysis and experiments are performed before the midblastula transition.

We have rephrased the text to clarify this issue:

“It is worth noting that the low assortativity scores we measure for GAF were obtained at nc14, before the midblastula transition, whilst previous studies reported GAF focal loops directly visible in Hi-C maps after midblastula transition^{30,32}. Low assortativity scores can arise when the presence of a factor is not associated with presence of loops (Fig. 1a), or if the factor is present only in a very small proportion of chromatin loops. Thus, we conclude that before the midblastula transition, GAF is not associated with chromatin loops.”

1.6 - At the end of the section “Genomic regions displaying preferential interactions are predominantly bound by chromatin insulators”, it is stated that “ChAs do not shed light on whether and how binding of insulator may modulate interaction frequencies”. However, I believe that the ChAs values do indeed reflect the interaction frequency of the insulator. This is because significant interactions are used to construct the network, so it seems logical to me that high ChAs should be linked to high $\log_2(O/E)$ values. Figure 1c demonstrates that these two quantities are strongly correlated and effectively convey the same information. Could the authors provide their thoughts on this?

That sentence was to highlight that high ChAs does not imply causality in any way, but we agree that it is confusing. Indeed, ChAs and APA provide overlapping, but also complementary information. On one hand, ChAs analysis is performed specifically on the network of chromatin loops. On the other hand, APA measures the interaction enhancement with respect to the expected interaction for every insulator peak (irrespective of whether it is or not involved in a chromatin loop). Thus, we expect the results from both methods to highlight which insulators are most relevant for the formation of chromatin loops, and as expected we observe a strong correlation between the two analysis methods. To clarify this issue, we removed the offending sentence and now conclude the section as follows:

“...Overall, these results indicate that different *Drosophila* insulators tend to be associated with 3D interaction networks to different degrees.”

1.7 - Line 174-176, it states that preferential interactions involved borders are minority of all preferential interactions, 1% for border/border etc. It would be intriguing to see the impact of normalizing the numbers of borders and non-borders on these percentages. Given that the number of borders is likely much smaller than that of non-borders, the percentage of border/border interactions would be greatly impacted. Normalizing with respect to their numbers could shed light on whether the likelihood of border/border interaction is higher or lower than expected.

Indeed, the number of borders is considerably smaller than the number of non-borders. Thus, from the data presented in the original submission, it was unclear whether border/border interactions occur more frequently than non-border interactions if their numbers were comparable. We addressed this comment by calculating the proportion of borders involved in loops called by Chromosight. We then stratified this calculation for loops containing borders in one or both anchors (see panels S2c and S2d below). We found that ~38% of borders took part in loops, with the majority of those (36.8%) being involved only in one loop anchor. To compare this to the probability with which non-borders participate in loops, we calculated similar statistics for all members of the Class I IBPs group: BEAF-32, Chromator, DREF, Z4, ZIPIC, and ZW5 peaks. For this, we removed peaks that overlapped with borders. We find that the propensity of non-border insulator peaks to form loops is lower than that of TAD borders.

We added these new panels and modified the text as follows:

“The number of loop anchors corresponding to TAD borders is considerably larger than the number of non-borders. Thus, we estimated the probability with which a border may take part in a loop by calculating the proportion of borders participating in loops (either in one or both anchors). We found that ~38% of borders take part in loops in our Chromosight network (Fig. S2c), with the majority of them participating as a single anchor (~36.8%) (Fig. S2d). Next, we calculated similar statistics for Class I IBPs non overlapping with borders. Notably, we found that the propensity of non-border IBP peaks to form loops was always lower than that of TAD borders (Figs. S2c-d). Overall, these results are consistent with Class I IBPs binding at loci displaying preferential looping, at both border and non-border regions.”

1.8 - Line 205, “ensemble pairwise distance maps were built by kernel density estimation of the full pairwise distance”. Since the ensemble pairwise distance maps are simply the average distances map, I'm curious as to why the authors opted for using kernel density estimation instead of simply calculating the mean values of the distances. The ensemble pairwise distance maps appear to be an average of the distances, so what additional benefits does the kernel density estimation provide?

We thank the reviewer for raising this point. We provide a side-by-side comparison of the maximum of the kernel density estimation (KDE) and the median method used to compute the ensemble pairwise distance Hi-M maps. Both methods provide very similar matrices, which show a high degree of correlation (see Table below, Pearson correlation = 0.93). KDE is a method that is more suited for noisy PWD distributions, which is not the case at hand. Thus, we agree that KDE may unnecessarily complicate the analysis and now follow the reviewer advice and use median matrices instead.

The maps in Figures 3 and 4 were updated, and the text was amended as follows:

“Ensemble pairwise distance maps were built by calculating the median of the full pairwise distance (PWD) distributions (**Fig. S3e**).”

1.9 - When I compare the average distance map in Fig. 3c to the Hi-C map in Fig. 3b, it seems like the average distance map has much more noise and the TAD structures are not as clear as in the Hi-C map. Have the authors calculated the standard error for the average distances? How many samples were used in the calculation and what level of confidence can be placed on the interactions marked by the yellow and green arrows in Fig. 3c?

Motivated by this comment, we reanalyzed our HiM dataset using an analysis that takes into account local distortions between imaging cycles and thus improves the drift correction. The application of this method leads to an improvement in the Hi-M matrices: we now see the TADs better and the correlation with the HiC matrix improves from 0.84 (contact threshold of 250 nm) in the original manuscript, to 0.91 (200 nm contact threshold) in the revised version. Importantly, the main interpretation/conclusions remain unchanged. We have revised the panels as follows:

The revised Hi-M distance map is less noisy than the previous one and TAD structures are considerably clearer:

To test whether these statistics were sufficient to reconstruct a consistent map, we estimated the number of Hi-M traces needed to reach a high correlation with the ensemble matrix (that contains all the available statistics) using a bootstrapping approach with 250 randomizations. We found that ~500 traces are needed to reach a Pearson correlation of 0.9 and ~1500 traces to reach a Person correlation of 0.98. We have added this analysis to Supplementary Fig. S3i. The Hi-M maps presented in the revised manuscript are built from a total of 23,531 traces from 22 embryos for nc14, and 1792 traces from 4 embryos for nc12. Two biological replicates were performed per condition. Thus, the number of traces acquired was sufficient to obtain an ensemble matrix that does not change anymore.

We have modified the *Image processing* methods section to include the statistics used to reconstruct the matrices.

“From the list of pairwise distance maps, we calculated the proximity frequencies as the number of nuclei in which pairwise distances were within 200 nm normalized by the number of nuclei containing both barcodes. Hi-M maps of nc14 embryos were generated from a total of 23531 traces from 22 embryos from 2 separate experiments. The maps for nc12 embryos are constructed from 1792 traces from 4 embryos from 2 separate experiments.”

We have also removed the yellow and green arrows in Figure 3c.

1.10 - One of the findings in the study is the absence of rosette-like structures among insulators, due to the low probability of having multiple contacts between insulators. I concur with this conclusion as the imaging data in this study, as well as previous studies, show that at any given cell, the probability of multiple loci being close together is low, which is understandable as the probability of having multi-way contacts decreases when pairwise interactions are not strongly dependent. But, I think a different angle could be considered as well. Is the probability of multi-way contact higher or lower than expected? This idea is similar to the Observed/Expected analysis commonly performed in Hi-C data. By computing the multi-way (at least three-way) interactions from imaging data, a more informative result could be obtained, and potentially highlight "significant" multi-way contacts and interactions.

We agree with the reviewer that the probability of multiple loci interacting is very low. To ask whether the observed values are significantly higher or lower than what is expected we calculated the probability of clustering assuming that these events are independent.

Assuming that :

A is interacting with B in x% of the cells.

B is interacting with C in y% of the cells.

C is interacting with A in z% of the cells.

For instance, the expected 3-way clustering probability is $P(x) \cdot P(y) \cdot P(z)$. We computed these probabilities for all types of multi-way interactions and found that the observed contact frequency is always higher than the expected values, implying that even if these multi-way contacts are rare, they happen more often than expected (see figure below). We note, however, that proximity frequencies may depend on chromatin state and bound factors,

which is not considered in this simple calculation. This panel now appears as Fig. 3f in the revised manuscript.

The results show that experimental multi-way interactions are rare, but still slightly higher than those expected by chance. We added this conclusion to the revised text :

“The frequency of multi-way interactions rapidly decreased with the number of co-localizing targets but is still higher than what would be expected by chance (Fig. 3f, see *Multi-way proximity frequency analysis Methods*).”

We also amended the *Multi-way proximity frequency analysis* section methods by incorporating a detailed explanation of the calculation procedure:

“The expected proximity frequency is derived by considering all events as independent. Briefly, we computed the mean of the product of all possible barcode combinations for various numbers of interacting partners.”

1.11 - Regarding the statement on line 264-266 about the overall structures of the *dpp* locus considerably changing, it would be helpful to see a different comparison or plot for better visualization.

To address this point we have now included in Fig. 4b the reference plot for *nc14* and *nc12* as well as the difference between the two in Fig. 4c. This new figure now appears in the revised manuscript.

1.12 - Line 269-271 states that insulator colocalization is present as early as *nc12*, but when I look at Fig.4b, it seems to suggest otherwise. The proximity probability of insulators and non-insulators is almost the same (12.5% vs. 10.9%). This raises questions about whether these interactions are actually stabilized at *nc12* or if they just happen due to polymer fluctuations. Can the authors provide clarification on this issue?

We agree with the reviewer on this point, and have removed this conclusion accordingly.

1.13 - On line 286-288, it states that “the increase in interactions between non-border insulator-bound regions is consistent with enhanced inter-TAD interaction”. To properly support this statement, it would be necessary to first separate the interactions between non-border insulators into inter-TAD and intra-TAD, and then focus only on the inter-TAD interactions among non-border insulators.

We agree with the reviewer and thus addressed this point by quantifying the percentage of inter-TADs loops in nc14 wt and nc14 embryos treated with inhibitors of RNA Pol II activity (Triptolide and alpha amanitin). The results of our analysis demonstrate that inter-TAD interactions are indeed enhanced when RNA Pol II is inhibited. We have included this new analysis in our revised manuscript in Fig. S4f (see below), to provide further justification for this statement.

The text was revised as follows to make reference to the newly added figure panel in Fig. S4f :

“The increase in interactions between non-border insulator-bound regions is consistent with enhanced inter-TAD interactions ²⁰ (Fig. S4f)”

1.14 - The resolution of figures in the SI is too low. For some of the figures, the texts and numbers are unreadable.

We thank the reviewer for pointing this out. We have changed the illustration and the font text in all the supplement panels to correct this issue.

1.15 - The x-axis label for Fig. 1C should be “mean of log₂(O/E)” or something like that

We thank the reviewer for raising this issue. We have changed the x-axis label and the misleading colors, see 3.8.

1.16 - XYZ coordinates data from the imaging experiment should be provided.

We agree with the reviewer. We uploaded XYZ coordinates to our OSF repository and adapted the section on data availability in the text as follows.

“Single nucleus pairwise distance matrices as well as XYZ coordinates of chromatin traces generated in this study were deposited at our [Open Science Framework project](https://openframeworkproject.org/) with DOI: [10.17605/OSF.IO/AQTXJ](https://doi.org/10.17605/OSF.IO/AQTXJ). The list of previously published datasets used in this study is provided in Supplementary Table 1.”

1.17 - The details of how the insulation score (IS) is calculated should be provided in the methods section or SI

The insulation score metric is calculated in Hug et al. 2017 according to the definition by Crane et al., 2015 on the 5-kb contact matrices with a window size of 8 bins. We have added this information to the Methods section, as follows:

“Boundary calling

A previously annotated list of TAD boundaries was used in this study. Briefly, boundaries were called using the insulation score metric defined by Crane et al., 2015 using a 5kb balanced contact matrix with a window size of 8 bins.”

1.18 - The distribution of log₂(O/E) shown in Fig. S1c seems to be multi-modal. Does this indicate that there may be sub-classes of a given IBP?

We thank the reviewer for pointing this out. We reasoned that multi-modality could arise from different genomic distances between insulator-bound genomic loci. Thus, we re-calculated these distributions by applying a maximum distance threshold of 250kb, which is considerably larger than the average TAD size in *Drosophila* (~70kb). The distributions of log₂(O/E) for these mid-range distances follow a monomodal distribution (see figure below).

Thus, we conclude that the second peak at negative $\log_2(O/E)$ values observed in the original Fig. S1c were mainly due to interactions between distant insulator sites. Similar conclusions can be reached for other factors (Rad21, Pc, Ph S5P and Zelda). To clarify this point, we have added an additional panel (Fig. S1k) showing the distribution of $\log_2(O/E)$ for all factors using a maximum distance threshold of 250kb.

We also modified the original panel Fig. S1c to show the two different ‘peaks’ on the distribution see below:

We amended the text as follows:

“Remarkably, most of the insulator factors displaying positive ChAs Z-Scores also exhibited positive $\log_2(O/E)$ (BEAF-32, CHRO, DREF, Z4, ZIPIC and Zw5) (hereafter referred to as Class I insulators) (Figs. 1b, S1j). The peaks observed for negative $\log_2(O/E)$ values (referred as *peak 2* in Fig. S1h) are related to long-range contacts (Fig. S1j). Consequently, it can be inferred that Class I insulator sites exhibit a higher tendency to interact with each other at shorter distances (<250kb, Fig. S1k-I).”

Reviewer #2 (Remarks to the Author):

This manuscript by Messina et al. is addressing an important question about the roles of insulator proteins in Drosophila. The combination of computational analysis of chromatin assortativity scores with in vivo Hi-M has the potential to provide significant insights into this topic. However, the chromatin assortativity analysis lacks sufficient quality controls. In addition, there are several points where the authors overstate their findings or where the phrasing is unclear. Overall, the paper requires additional analyses to clarify and confirm the robustness of their findings.

We thank the reviewer for their thorough review of the manuscript and their suggestions for improvement. We have taken the reviewer's suggestions into account and made appropriate revisions to the manuscript and figures, which are outlined below.

Major points

2.1 - The Chromatin Assortativity analysis relies on the input network being robust and reliable. Here, the input network is based on loop calls by Chromosight, but no analysis of the quality or robustness of these loop calls is performed. This is particularly important since Drosophila loops have different characteristics from mammalian loops (e.g., they are not typically found at domain corners), which affect the ability of loop callers designed for mammalian data to detect them. Chromosight does not appear to have been tested on Drosophila data in the original publication. Therefore, to support the use of this network for downstream analysis, it is essential that the authors include an analysis of the quality and robustness of the loop calls used for the network, including typical loop sizes and how well their loop calls overlap with those from other studies (e.g. Batut et al. 2022, PMID: 35113722), as well as visualisations of example loci with loops.

The reviewer is correct in that Chromosight was initially trained to detect patterns in various genomes, including bacteria, viruses, yeasts and mammals. To validate the capacity of chromosight to correctly identify loops in *Drosophila* we conducted a benchmarking study to evaluate its performance on *Drosophila* data. As indicated in reviewer response 1.2, we set the genomic distance range for loop detection between 20kb and 2Mb, and the Pearson correlation threshold used to define significant interactions to 0.3. With this set of parameters, we detected 2153 loops from 3874 non-overlapping anchors. The loop size distribution is now shown in the revised Fig. S1c:

As an output, Chromosight produces a pileup plot of all detected loops. The plot shows the average signal from 2153 detected loops. The resulting pattern matches the expected

pattern with a strong signal (loop score) at the center and a decreasing score as one moves away from the center. We added this panel to Fig. S1b:

To give a more visual representation of loops detected at single loci we also represent a few examples of loops detected by Chromosight that can be directly seen on the Hi-C data. These examples are now shown in Fig. S1d of the revised manuscript. We note that the matrices displayed below are not normalized by genomic distance, while Chromosight performs this normalization to detect loops.

Using standard parameters, Chromosight detected 3874 loop anchors when applied to the nc14 data from Hug *et al.* (2017), which is considerably higher than the number of loops reported by Batut *et al.* 2022 in their microC dataset (614 loop anchors). Chromosight detects regions interacting preferentially, and a large proportion of interactions do not appear as discrete focal loops (see examples above). In contrast, Batut *et al.* applied more stringent loop calling algorithms to detect specifically focal loops. This explains why the number of loops detected by Chromosight on Hug’s data is higher than the number reported by Batut *et al.* (2022) that. This said, most loops reported by Batut *et al.* (71.33%) were also detected by Chromosight applied to Hug’s dataset, providing further support to our loop calling approach (see Table below).

To further benchmark Chromosight, we ran it on the micro-C data of Batut *et al.* and found that 78.33% of the loop anchors detected by Chromosight on Batut's data overlap with the loop anchors reported by Batut himself. These results provide additional support to the validity of our approach. These results are shown in the Table below:

	Chromosight applied to Hug et al. (2017) data	Chromosight applied to Batut et al. (2022) data
Loops reported by Batut et al. (2022).	a Loop anchors Chromosight Hug et al. 2017 Loop anchors Batut et al. 2022 3436 438 (71.33%) 176 Additional panel: Fig. S1a.	4205 481 133 Batut et al. 2022 Chromosight
Overlap with Batut et al. (2022).	71.33 %	78.33 %

The robustness of Chromosight was further tested in response to a request from Reviewer 1 (see 1.2).

2.2 - The insulator ChIP-seq datasets used are from cell lines, while the Hi-C data is from early embryos. The limitations of the data should be discussed. E.g. while interactions between regions bound by these insulators appear before TADs, it's not clear whether these IBPs are actually bound at nc12. Additional datasets from embryos and/or analysis of chromatin accessibility data at nc12 would support the idea that the IBPs actually bind to these loci at this timepoint.

We agree with the reviewer in that the data currently available does not demonstrate that class I IBPs are bound at these early nuclear cycles, as our analysis mostly relies on ChIP-seq data from cell lines. Thus, we added a comment to the results section as follows:

“Notably, we found that preferential interactions between non-border regions bound by insulators were already present in nc12 embryos for most Class I IBPs (Figs. 4c, 4d). We note, however, that further studies will be required to fully establish whether these sites are actually bound by Class I IBPs at these early stages of development.”

Next, we followed the advice of the reviewer and used the ATAC-seq dataset from Blythe *et al.* *Elife* (2016), where they reported accessibility for different developmental times, to ask whether class I IBPs sites are open as early as nc12.

For this, we downloaded and analyzed .wig files from GSE83851 and converted them to .bw using *wigToBigWig*. Next, we used *computeMatrix* and *plotHeatmap* from *deeptool* v3.3.0 to compute ATAC-seq signals on identified IBPs regions over a window of +/- 1kb for each developmental stage including nc12 (3'/6'/9'/12'), and nc13 (3'/6'/9'/12'/15'/18'). These analyses are summarized in the new Fig. 4a of the revised manuscript:

Additional panel: Fig. 4a

These data show that a large proportion of IBPs sites are accessible as early as nc12, with accessibility increasing during development (between nc12 and nc13). To directly visualize the average accessibility profiles of the regions occupied by each member of the class I IBPs groups, we separated each protein and calculated the average ATACseq signal over a +/- 1kb window (panel Fig. S4a below):

Additional figure now in Fig. S4a

This analysis more quantitatively shows that accessibility increases during development from nc12 to nc13. We note that these new analyses do not demonstrate that insulators are bound to these sites at nc12. However, they show that at these developmental times these sites are accessible and preferentially interact. We can thus only infer that insulators may play a role in these preferential interactions at this early developmental stage.

We have considerably revised the text of this section of the manuscript to incorporate these results and discussions (see pages 8-9, major changes highlighted in blue). We also updated the methods section to explain the methodology used for these new analyses:

“ATAC-Seq data processing

ATAC-seq data were downloaded from GSE83851 (Blythe and Wieschaus 2016). *Wig* files were converted to *BigWig* using *wigToBigWig* from UCSC. Heatmaps of ATAC-seq profiles were then plotted over +/- 1kb window centered on Class I IBPs sites using *computeMatrix* followed by *plotProfiles* from *deepTools* (Ramírez et al. 2016). Average ATAC-seq profiles derived from the heatmaps for individual IBPs were constructed using a custom Matlab script.”

2.3 - None of the factors tested have a negative ChAs score – are any chromatin proteins expected to be disassortative? How was the threshold of 2 chosen to select those for further analysis?

→ Disassortative proteins expected ?

In ChAs analyses, proteins may have a disassortative score and especially histone modifications clearly do in certain cases. However, this is usually dependent on the DNA fragment annotations that make up the network. For example, ChAs calculations on networks constructed using interactions between gene promoters and distal regulatory elements will show disassortative scores for proteins and histone marks that are specific to either promoters or to regulatory regions (see Pancaldi et al. 2016). This is not the case here, as our network was constructed using DNA fragments without specific genomic annotation.

None of the proteins we considered in this study are expected to be disassortative. On this specific network, we might have found disassortative proteins in the case in which loops are associated to the presence of protein A on one side and protein B on the other side while loops are never found when both sides have protein A or both have protein B (hypothetically one could imagine this happening if A and B form a complex that mediates the loop).

→ ChAs Z-Scores threshold of 2 ?

We are aware that the choice of a Z-Score can be a tricky field and is not always necessary. In our case, ChAs Z-Scores are calculated to represent the standard deviation between the protein ChAs value and the ChAs distribution in the randomisations. Given that the randomized ChAs values tend to be normally- or at least mostly monomodally-distributed (see figure - Distribution of ChAs from randomizations below), a ChAs Z-Score over the threshold of 2 establishes that the measured protein ChAs is higher than at least 95% of the randomized ChAs values. This threshold corresponds to a significant p-value (0.05) and it helps us to categorize proteins with a ChAs Z-Score > 2 as highly assortative.

It must be pointed out that randomisations here take into account the distance spanned by interactions. So the Z-Score actually measures whether the preferential contacts are more than expected given the correlation of the features along the genome. For example, if a chromatin feature spans multiple fragments (along the genome) we will definitely see high ChAs values (since genomically close regions will also interact in 3D) but, unless this feature is related to the presence of 3D interactions, we will not have a significant Z-Score.

2.4 - The aggregate peak analysis has been carried out using all pairs of sites – is there a difference if using only pairs within a certain distance? It would be interesting to see if some factors have preferential interactions only at short distances, which are not seen when including long-range pairs.

In line with the previous comment from reviewer 1 (see point 1.13 above), we have computed the $\log_2(O/E)$ distribution for all factors (as previously in Fig. S1) but only considering genomic distances shorter than 250 kb. The original and new panels are shown below:

The distributions of $\log_2(O/E)$ at genomic distances <250 kb follow a monomodal distribution (see right panel, now in Fig. S1k of the revised manuscript). Thus, we conclude that the second peak at negative $\log_2(O/E)$ values observed in the original Fig. S1c (left panel) were mainly due to interactions between distant insulator sites. Similar conclusions can be reached for other factors (Rad21, Pc, Ph, RNAP-S5P and Zelda). This analysis also shows that factors tend to have more interactions at shorter genomic distances, which is expected. However, the overall results remain the same.

We also computed aggregate Hi-C plots for Class I IBPs factors with a maximum distance cutoff of 250 kb (see figure below). Consistently with the panel above, this new plot suggests that Class I IBPs sites tend to interact more at shorter distances.

We amended the text as follows:

“Notably, the positive correlation between ChAs and $\log_2(O/E)$ (**Fig. 1c**) indicates that factors displaying high assortativities are bound to chromatin regions that exhibit the most preferential interactions. Remarkably, most of the insulator factors displaying positive ChAs Z-Scores also exhibited positive $\log_2(O/E)$ (BEAF-32, CHRO, DREF, Z4, ZIPIC and Zw5) (hereafter referred to as Class I insulators) (**Figs. 1b, S1j**). The peaks observed for negative $\log_2(O/E)$ values (referred as *peak 2* in **Fig. S1h**) are related to long-range contacts (**Fig. S1j**). Consequently, it can be inferred that Class I insulator sites exhibit a higher tendency to interact with each other at shorter distances (<250kb, **Fig. S1k-l**).”

2.5 - To better link the two halves of the paper, it would be helpful to show the loops, if any, in the *dpp* locus, that were called with ChromSight and used for the network analysis.

To address this question, we used ChromSight to call loops at the *dpp* locus (network 1, see 3.1 below). We found 6 intra-locus loops, and 18 loops containing at least one anchor within the *dpp* locus. We added this panel to the revised manuscript (Fig. S3a).

We modified the text as follows:

“Specifically, we imaged the 3D chromatin organization of the *dpp* locus (*chr2L*: 2343645-2758688 dm6) in intact nc14 *Drosophila* embryos at ~12 kb resolution (**Figs. 3a, S3a**). The *dpp* locus contains three TADs, multiple preferential loops (**Fig. S3a**), and several regions displaying high levels of class I insulator binding, named barcode I1 to I10 (**Fig. 3b**).”

2.6 - The authors conclude that multi-way interactions are rare. Would it be possible to calculate how often triplet/quadruplet/etc interactions would be expected by chance, given the observed pairwise interaction frequencies? It would be informative to know if multiway interactions occur more, less, or equally as often as expected by chance.

We agree with the reviewer and have computed the expected contact frequency expected by chance for different numbers of interacting partners (see analysis reported in 1.10 above). This new panel now appears as **Fig. 3e** in the revised manuscript.

The results show that experimental multi-way interactions are rare, but still slightly higher than those expected by chance. We added this new panel and conclusion to the revised ms:

“The frequency of multi-way interactions rapidly decreased with the number of co-localizing targets but is still higher than what would be expected by chance (Fig. 3f, see *Multi-way proximity frequency analysis Methods*).”

We also amended the methods by incorporating a detailed explanation of the calculation procedure :

“The expected proximity frequency is derived by considering all events as independent. Briefly, we computed the mean of the product of all possible barcode combinations for various numbers of interacting partners.”

2.7 - The introduction states that *Drosophila* insulators “do not seem to rely on loop extrusion”, yet this doesn’t follow from the previous paragraphs – loop extrusion itself can occur without CTCF-mediated blocking. The authors should rephrase or clarify this point.

The reviewer is correct. We removed the offending sentence. The text now reads as follows:

“Early genome-wide studies showed that insulators preferentially bind to genomic regions containing housekeeping genes and highly transcribed regions¹⁶. In addition, IBPs frequently bind to TADs borders^{4,6,17–19} that can often interact in 3D²⁰. Taken together, these data suggest that insulators may be involved in the organization of *Drosophila* TADs, ~~yet do not seem to rely on loop extrusion.~~”

2.8 - The explanation of how low assortativity can arise (lines 116-118) is unclear – a schematic might help. Because of this I find that the justification for low assortativity of GAF is not well-explained. The authors should clarify this part of the text. As mentioned above, it would also be very informative to check the overlap of their loop calls with the GAF-bound tethers identified by Batut et al.

Assortativity refers to the tendency of nodes with similar characteristics (i.e. bound by the same factor) to connect with each other in a network. High assortativity means that nodes bound by a particular factor tend to be highly connected, forming a cohesive subnetwork. Disassortativity occurs when nodes bound by a factor preferentially interact with nodes NOT bound by the same factor. Low assortativity is an intermediate state where there are no preferential interactions with either bound or unbound nodes.

We now illustrate this by modifying Figure 1a to include a schematic representation of the low assortativity state:

And the text as follows:

“Low assortativity scores can arise when the presence of a factor is not associated with presence of loops (Fig. 1a), or if the factor is present only in a very small proportion of chromatin loops.”

Regarding the last part of the question, as mentioned above, we compared the overlap between the loop anchors identified in Batut et al. 2022 with the loop anchors detected by Chromosight in our study and found a large degree of overlap between the two (see 2.1 above). Several new panels were introduced in the revised text to present these new results.

Tethers are characterized by their association with pioneer factors such as Trithorax-like (Trl), grainyhead (grh), and zelda (zelda), along with the presence of H3K4me1. In our analyses, GAF exhibits both negative $\log_2(O/E)$ values and low ChAs Z-scores, which seems to contradict the previous findings (Ogiyama et al., 2018 and Loubiere et al., 2020) showing that GAF is important for the formation of repressive loops. This apparent discrepancy can be explained by the fact that repressive loops involving GAF appear after the midblastula transition (Ogiyama et al., 2018), whereas our analysis and experiments are performed before the midblastula transition. The text was adapted to explain this point (see also 1.5 above):

“It is worth noting that the low assortativity scores we measure for GAF have been obtained at nc14, before the midblastula transition, whilst previous studies report GAF focal loops directly visible in Hi-C maps after midblastula transition^{29,31}. Low assortativity scores can arise when the presence of a factor is not associated with presence of loops (Fig. 1a), or if the factor is present only in a very small proportion of chromatin loops. Thus, we conclude that before the midblastula transition, GAF is not associated with chromatin loops. ...”

2.9 - Line 132-134: the current phrasing implies that chromatin factors with positive ChAs scores cause increased interaction strength, however the data presented cannot show causation. Please rephrase this sentence.

We agree with the reviewer comment regarding the potential ambiguity in our phrasing. To disambiguate, we revised the text as follows :

“Notably, the positive correlation between ChAs and $\log_2(O/E)$ (**Fig. 1c**) indicates that factors displaying high assortativities are bound to chromatin regions that exhibit the most preferential interactions.”

2.10 - Fig 2a: It's unclear to me whether this a schematic or shows the actual network. If this shows the actual network, it doesn't seem to be consistent with the very low % of border-border interactions in Fig 2e. The authors should clarify this.

We are sorry for the confusion. Fig. 2A is not a scheme but rather showed pieces from the total network and not the entire network itself. Our goal was to illustrate how data was managed and represented as visual support. Those network parts were chosen to highlight that TAD borders are highly assortative. Below we display a global view of the network from which parts from Fig. 2a were taken (subnetworks composed of two nodes only have been masked). This new panel is included as a supplementary figure:

In addition, we have changed the figure legend to clarify this point:

“a. Chromosight chromatin subnetwork from Hi-C data at nc14 embryos²⁰. Each node of the network is a chromatin fragment, blue nodes represent nodes in which a TAD border is found, and edges represent significant 3D interactions.”

2.11 - Lines 186-187: the strong interaction seen in Fig 2f does not imply anything about the binding of IBPs, but rather is simply consistent with the aggregate analysis of called loops. This sentence should be rephrased to clarify what the authors want to show here, or the analysis should be removed.

We apologize for the lack of clarity in the original submission. Fig 2f shows aggregate analysis of regions bound by class I IBPs rather than aggregation of called loops. To disambiguate, we modified the text as follows:

“To further support this conclusion, we performed aggregation Hi-C analysis on non-border/non-border regions occupied by Class I IBPs. Notably, this analysis displays a

clear peak (**Fig. 2f**), suggesting preferential interactions between anchors containing Class I IBP sites.”

2.12 - I think line 231 has a typo - non-insulator barcodes colocalise at a slightly lower frequency than insulator barcodes.

We reworded this section to include new analysis, this does not need correction any longer.

2.13 - I find it hard to identify/appreciate the key differences in Fig 4a. It would be helpful to show the reference plots for nc14 and nc12 as well.

To address this point we included in Figure 4 the reference plot for nc14 and nc12 as well as the difference between the two. This new figure now appears in Fig 4e (left) and Fig 4f (right) of the revised manuscript.

2.14 - Line 269 – what is meant by “specificity” here? How is this measured?

We apologize for the misunderstanding. Rewording of the text to incorporate new analysis lead to changes in this sentence which does not refer any longer to ‘specificity’.

2.15 - Line 318 – the wording “class I insulator family” might inadvertently imply an evolutionary relationship between these proteins.

To avoid ambiguity, we changed the text as follows:

“Members of the class I insulator **group** (e.g. BEAF-32) tend to co-localize with promoter regions”

Reviewer #3 (Remarks to the Author):

In the manuscript “3D chromatin interactions involving Drosophila insulators are infrequent but preferential and arise before TADs and transcription” Messina and colleagues combined bioinformatics analysis and multiplexed chromatin imaging to unveil the contribution of insulators in shaping the 3D genome organization in Drosophila, that intriguingly differ from mammals one in varied aspect. Characterizing these differences better is an interesting and fundamental question in the field that, interestingly, is also accumulating provocative and contradicting findings. Generally, the presented data are, for the most part, convincing, although in some sections, the manuscript lacks the necessary clarity to convince a reader that the results are robust and relevant. Additionally, I recommend toning down the title and making it less generic, as the work presented supported by HiM experiment focuses on a small genomic region spanning only 3 TADs. To this end, I have some concerns that need to be improved before publication.

We thank the reviewer for their comprehensive evaluation of the manuscript and their valuable feedback. We have carefully considered the reviewer's suggestions and incorporated them into the revised version of the manuscript and figures. A summary of the changes made and point-by-point response is provided below.

Specific critiques need to be addressed and discussed:

3.1 - *One of my concerns is related to the first section of the results, where the authors dissect the preferential interactions of a set of 15 possible insulator proteins using Chromatin Assortativity analysis (ChAs). The analysis, by design, creates a chromatin interaction network focusing only on highly frequent interactions present in the population ensemble experiment (HiC data). This network construction can compromise the results and conclusions as it does not consider the multiple low-frequency and low-affinity interactions that have a role in TAD-like structure generation (as the authors identify in previous work (Cattoni Nat Commun 2017)). Specifically, I found interesting the identification of class I and II insulators; however, it would be nice to see if this classification and the Assortativity of the network hold when also the low-frequency interactions are accounted for.*

We thank the reviewer for raising this point. To test whether the relative assortativities of IBPs changes when lower-frequency interactions are considered, we generated two additional networks with by varying Chromosight's parameters to increase the network size (see table below). Increased network sizes contain more low-frequency interactions.

	Set 1 (Network 1) Set used in the ms	Set 2 (Network 2)	Set 3 (Network 3)
Parameters used	--pearson 0.3 --min-dist 20kb --max-dist 2Mb --min-sep 5kb --max_perc_0 10	--pearson 0.3 --min-dist 10kb --max-dist 200Mb --min-sep 5kb --max_perc_0 50	--pearson 0.2 --min-dist 10kb --max-dist 200Mb --min-sep 5kb --max_perc_0 50
Number of loops called	2153	3206 (+ 48.91 %)	17567 (715.93 %)

Next we calculated the ChAs Z-score for each set of parameters and compared the results obtained for each network. We found that the ChAs scores from these new networks correlated very well with those of our original, more restrictive network (network 1). Thus, the incorporation of lower-frequency interactions does not affect our general conclusions.

We revised the text as follows:

“To validate the robustness of these results, we performed similar analysis for different sets of ChromSight parameters (see chromatin assortativity Methods) generating larger networks that include lower-frequency interactions. We found that the insulator factors with the highest ChAs Z-scores were the same independently of the size of the network (Fig. S1f).”

3.2 - Another aspect that has not been investigated thoroughly in this analysis but that I believe needs to be addressed to truly dissect the insulators' role in the folding of the zygotic genome during early embryogenesis genome-wide is considering the combination of these factors. Indeed, it has been shown, for example, that the depletion of the insulator BEAF-32 does not abolish boundaries (Ramirez et al., Nat. Commun. 2018 PMID: 29335486), possibly suggesting that potentially a combination of factors is required for boundary maintenance. The author tried to address this aspect by HiM co-localization analysis (see below) but only on a small locus of ~415kb located in chr2L:2343645-2758688 and not fully genome-wide. How does the Chromatin Assortativity analysis change if the combination of class I and/or class II insulator are considered?

We thank the reviewer for asking this relevant question. We actually can address it at the genome-wide scale by using two other types of assortativity analyses: CrossChAs and AND-ChAs (Madrid-Mencia et al, 2020). On the one hand, CrossChAs measures assortativity of nodes bound by two different proteins, and gives information about frequency of interactions joining fragments with one protein on either side. On the other hand, AND-ChAs measures preferential interactions between nodes that are both bound by a pair of factors, and therefore provides information about interaction frequencies of co-localized proteins.

The figure below displays the CrossChAs and AND-ChAs Z-Score heatmaps for the factors investigated in this study, in nuclear cycle 14 embryos. The CrossChAs heatmap shows that a subset of IBPs (BEAF-32, Chromator, Z4, PolII, Zelda, L3(MBT), DREF) display high cross-assortativities, suggesting that anchors bound by combinations of these factors tend to preferentially interact together. The AND-ChAs heatmap shows that DNA fragments containing colocalized factors (BEAF-32, Chromator, Z4, PolII, Zelda, L3(MBT), ZIPIC) show preferential interactions with fragments containing the same pair of factors. Thus, pairs of IBPs can be found at each anchor of strong loops. The set of insulators that display the highest Cross-ChAs and AND-ChAs scores correspond to the class I insulators identified in this manuscript.

This analysis is consistent with class I IBP's often acting together to promote long-range chromatin interactions, as suggested by the reviewer. We have included this analysis in the revised manuscript (Fig. S4g-h) and described the results as follows:

To investigate whether IBPs act together to promote preferential chromatin interactions, we employed Cross-ChAs and AND-ChAs²⁷. Cross-ChAs measures assortativity of two different proteins, giving information about frequency of interactions joining fragments with one protein on either side. Instead, AND-ChAs measures assortativity of two different proteins considering that connected nodes are bound by a pair of factors, and therefore provides information about interaction frequencies of co-occupied regions. We computed Cross-ChAs and AND-ChAs Z-Scores for each pair of factors investigated previously (**Figs. S1g-h**). Cross-ChAs shows that class I insulators (BEAF-32, Chromator, Z4, PolII, Zelda, L3(MBT), DREF) tend to display high cross-assortativities, suggesting that anchors bound by either of these factors tend to preferentially interact. AND-ChAs shows that DNA fragments containing colocalized class I insulators (BEAF-32, Chromator, Z4, PolII, Zelda, L3(MBT), ZIPIC) interact preferentially with each other. Thus, pairs of class I IBPs can be found at each anchor of strong loops. These results are consistent with Class I IBPs often interacting together to promote formation of preferential chromatin contacts in nc14 embryos.

3.3 - In the “Insulator binding increases the strength of preferential long-range chromatin contacts” section, the author refers to long-range interaction; however, the aggregation peak analysis is reported for only 100kb regions around the anchor point, which is more related to short-range interaction rather than long-range (generally occurring at the Megabase scale). This is valid throughout the presented manuscript; as such, I recommend that the author rephrase the relative sections to enhance the general clarity of the findings.

We are sorry for the apparent confusion. The 100kb window in the aggregation peak analysis represents the region around each peak that was used for alignment. It does not mean that we considered only interactions below 100kb. In fact, this analysis represented genomic interactions between preferentially bound peaks spanning all intra-chromosomal distances. We apologize for this misunderstanding and have adapted the Methods section accordingly:

“Hi-C aggregate plots were performed using a homemade analysis pipeline developed in MATLAB Release R2019b (The MathWorks, Inc., Natick, United States). The distance-normalized sub-matrices over a window of 100kb surrounding the intersection between two anchored peaks were extracted. Finally, the aggregate plots were then created by averaging all of the sub-matrices together.”

3.4a As the author points out in the manuscript, HiC is an ensemble-based method that cannot inform on single-cell patterns of interactions, an aspect that is possible to investigate with chromatin tracing methods such as HiM. Indeed, to account for this limitation, the author specifically imaged a small locus of ~415kb located in chr2L:2343645-2758688 subdividing the region into 34 equally spaced barcodes. 3 TADs characterize the locus in the population cell experiment (unfortunately, the color-bar scale of Figure 3 B does not show it fully, see minor points section).

We agree with the reviewer that the Hi-M maps shown in the original manuscript Fig. 3c does not clearly reveal the three TADs of the locus, as they do in Hi-C. In response to requests from both reviewers 1 and 2, we have improved the analysis software to account for local distortions observed between imaging cycles. This new method improved the Hi-M matrix and the visualization of the TADs at the *dpp* locus:

3.4b Unfortunately, it is unclear if the border-containing-barcode regions have been defined by Insulation Score analysis of the pairwise matrix or proximity frequency matrix derived by HiM. If not, how do the insulation profile of the HiM-derived matrices and the HiC compare?

The boundary annotations in Figure 3b were derived from the insulation score (IS) calculated from the Hi-C data and originally reported by *Hug et al. (2017)*. To see if this boundary identification is consistent with Hi-M data at the *dpp* locus, we calculated the IS and domainogram based on the median distance matrix from Hi-M for different genomic distances (from 12kb to 72kb):

These analyses show that borders called from HiM data (maxima in the insulation score, and red regions in the domainogram) agree well with those called from HiC data (see vertical dashed lines) at the *dpp* locus. In addition, they show that most preferential interactions occur within TADs (blue regions in domainogram).

We modified the main text and methods to reflect these new analyses, as follows:

“The proximity and PWD distance maps revealed multiple regions displaying preferential 3D spatial proximity (**Fig 3c**). These mostly corresponded to the TADs called from Hi-C data (Figs. 3b-c, blue arrows) and from Hi-M proximity frequency maps (Fig. 3c, insulation score, and domainogram).”

Insulation score derived Hi-M dataset

Insulation scores derived from the Hi-M dataset were computed by moving an n-by-n square window along the diagonal of the median pairwise distance and summing the distances within this square. Domainogram were calculated by smoothing a matrix obtained by computing the IS with an increased window size (from 1-by-1 to 6-by-6) over the Hi-M matrix.

3.4c The author reported the efficiency of detection as a percentage, but it would be nice also to have explicit quantification of the number of nuclei/ traces analyzed.

The results in Fig. 3 (nc14) were derived from two biological replicates that produced a total of 23531 traces from 22 embryos. For Fig. 4 (nc12), we performed two replicates amounting to 1792 traces from 4 embryos. This is now described in the captions of the corresponding figures as well as in the methods section.

To test whether the number of traces were sufficient to reconstruct a consistent Hi-M map, we performed bootstrapping analysis to estimate the number of traces needed to reach a high correlation with the ensemble matrix (see Fig. S3i below). We found that ~500 traces are needed to reach a Pearson correlation of 0.9 and ~1500 traces to reach 0.98. Thus, we conclude that the statistics for both nuclear cycles were sufficient.

We have modified the methods section to include the statistics used to reconstruct the matrices:

“From the list of pairwise distance maps, we calculated the proximity frequencies as the number of nuclei in which pairwise distances were within 200 nm normalized by the number of nuclei containing both barcodes. Hi-M maps of nc14 embryos were generated from a total of 23531 traces from 22 embryos from 2 separate experiments. The maps for nc12 embryos are constructed from 1792 traces from 4 embryos from 2 separate experiments.”

We revised the text as follows:

“The number of traces acquired was sufficient to ensure a statistically representative ensemble map (Fig. S3i, see *Image processing* in Methods).”

3.4d *To this end, it is known (also by a previous work of the author) that the nuclei in nc14 embryos are characterized by homologs showing pervasive pairing and that varied elements, including insulator elements, play a role in pairing (e.g., Rowley & Corces Nat. Rev. Genet. 2018). The author should elaborate on this aspect that delineates one of the main differences between 3D genome organization in Drosophila and mammals.*

The reviewer is correct that *Drosophila* displays a large degree of homologue pairing, in contrast to mammalian chromosomes where homologues are typically unpaired. We revised the Discussion section as follows:

Drosophila homologous chromosomes are often paired, and several factors, including insulators, play a role in this process (Rowley and Corces, 2018), therefore long-range

contacts between insulators bound to different homologous chromosomes could also contribute to *cis*-regulation (Galouzis and Prud'homme 2021).

3.5 - One of the main aspects of HiM is that it quantifies the real co-localized at the single-cell level. The author chose ten regions as control (marked in yellow) to probe if the insulator-bound regions are spatially co-localized. How does the cumulative curve change if another set of ten regions is considered as a control? Why did the author select only those to use as a control? I believe the analysis could gain strength if different sets of control regions are considered. This is needed to confirm the robustness of the minimal differences observed in the comparison between *nc12* and *nc14* embryos Hi-M maps when TADs emerge. That is also the main message of the manuscript title, which I recommend reconsidering since the author investigates only a specific locus.

We addressed this remark by performing two different analyses. First, we repeated the analysis shown in the original panel but we averaged 5 sets of control (non-insulator) barcodes. This analysis shows that the difference between insulator and non-insulator barcodes is rather small (12.19 versus 10.8%), consistent with our original analysis, and indicating that insulator barcodes have similar co-localization frequencies than non-insulator barcodes.

This analysis can be still potentially biased by the genomic distribution of insulator and non-insulator groups, as proximity frequencies strongly depend on genomic distance. Thus, to address this possible bias, we performed a new analysis where we calculated the proximity frequency as a function of genomic distance for both insulator and non-insulator barcodes, using a cutoff distance of 200 nm.

These analyses indicate that, at the *dpp* locus, barcodes co-localize at similar frequencies irrespective of whether they contain insulators. The average co-localization frequency remained relatively low (~10-12%), and exhibited a clear dependence with genomic distance, as expected. Remarkably, the difference between insulator and non-insulator barcodes remained small for all genomic distances, consistent with our original observation that the frequency with which insulators spatially co-localized seems mainly driven by polymer fluctuations, at least at the *dpp* locus.

We implemented the following changes to the text:

“Next, we investigated the specificity of insulator barcode co-localizations by calculating the proximity frequency versus cutoff distance curve for non-insulator (control) barcodes located at similar genomic distances (Fig. S3j, black curve). For this, we averaged 5 sets of control barcodes. At a cutoff distance of 200 nm, control barcodes co-localized at similar frequencies than insulator barcodes (10.8% and 12.19%, respectively). Next, we calculated how proximity frequency depended on genomic distance for both insulator and control barcodes, using a fixed cutoff distance of 200 nm (Fig. 3d). This analysis revealed that, at least at the *dpp* locus, barcodes co-localize at similar frequencies irrespective of whether they contain insulators. Proximity frequencies dropped with genomic distance, as expected, but the difference between insulator and non-insulator barcodes remained small for all genomic distances. Overall, these results are consistent with insulators being involved in stabilizing long-range interactions that are already created by other mechanisms (e.g. polymer fluctuations or other factors that we did not investigate). These infrequent interactions may be either short-lived or occur in only a subset of cells.”

We agree that our HiM results are relevant to the *dpp* locus, however, the rest of the data reflect genome-wide behaviors, therefore we would prefer not to modify the title of the manuscript. We have, however, noted in the section presenting HiM results that these apply to the *dpp* locus:

“...This analysis revealed that, at least at the *dpp* locus, barcodes co-localize at similar frequencies irrespective of whether they contain insulators.”

Additional comments:

Generally, I recommend editing part of the test and checking for general consistency.

3.6 - P4 line 86. “a chromatin interaction network is built from a single genome-wide contact map”, I found the use of ‘single genome-wide contact map’ misleading as it refers to an ensemble experiment.

We agree with the reviewer and amend the text as follows :

“In ChAs analysis, a chromatin interaction network is built from ~~a~~ a single genome-wide contact map.”

3.7 - P11 lines 394-395, the sentence is missing a verb.

We correct the paragraph as follows :

“Oligopaint libraries were constructed as in previous studies. Briefly, each oligo had an homology region of 35-41 nt followed by a flap encoding a sequence complementary to the readout probes.”

3.8 - Figure 1C, green labels are unclear. Z4 appears to be of different green and class than BEAF32, while in Figure 1E are listed as the same class. From the legend’s explanation, L(3)MBT and Pita should be class II, but they are color-coded differently. Please clarify.

We changed the confusing green gradient in Fig. 1c. See comparison before the original and the revised figure below:

3.9 - Figure 2H and Supplementary Figure 2C, in figure legend 2H, the “h” is capital letter and not in agreement with the rest of the text.

We thank the reviewer for pointing out the error in the figure legend, which has been corrected in the new version of the manuscript.

...Aside from this, the plots are not clear enough, and the annotation is not visible. The authors could elaborate better on these plots and clarify the confusion between insulation

strength (based on the insulation score) and insulator binding strength (how the latter is defined?).

We apologize for the lack of clarity in these plots. This plot was used to represent the correlation between ChIP signal and the mean of the $\log_2(O/E)$ as in Loubiere *et al.* (2020). The fact that highly bound regions exhibited an increase in the mean of the $\log_2(O/E)$ suggests that the binding of class I IBPs increases the strength of the interaction. By insulator binding strength, we meant the ChIP intensity signal. To clarify, we changed "insulator binding strength" to "ChIP signal" in the text and in Fig. 2h, as follows:

"Consistently, interactions mediated by Class I IBPs at non-border regions increased with **ChIP intensity (binding strength)** (Figs. 2g, S2e) of both loop anchors (Figs. 2h, S2f)."

3.10 - Figure 3 legend is poorly described: in panel B, *Drosophila melanogaster* needs to be in italics, and in panel C, the explanation of what the arrows are is missing. Additionally, in Figure 3B the color-coded is misleading, and it is difficult to see the TADs border. Please adjust it.

We apologize for this. We italicized *Drosophila* and added explanations where missing. We also adjusted the color scale in Fig. 3b, to improve visualization of TAD borders:

We revised the legend for Figs. 3b-c which now reads:

“b. Top: nc14 Hi-C matrix along the *dpp* locus (2L:2343645-2758688) in *Drosophila melanogaster* (dm6). Purple and green represent high and low contact probabilities, respectively. Identified TADs borders from nc14 embryos²⁰ are represented by blue triangles. TADs are highlighted on the matrix with black dashed lines. Barcodes used for Hi-M sequential imaging are represented as boxes, with barcodes bound by Class I IBPs displayed in green. Bottom : CHIP-seq profiles for Class I IBPs (BEAF-32, Chromator, DREF, Z4, ZIPIC, Zw5) aligned with genomic coordinates and gene locations.

c. Top: Hi-M pairwise distance (PWD) matrix for nc14 embryos constructed from 23531 traces from 22 embryos. Red and blue represent low and high distances, respectively. Middle: insulation score derived from Hi-M data with different window sizes (1, 2 and 3 bins), and domainogram (see Methods). Bottom: proximity frequency matrix from nc14 embryos (cutoff distance: 200 nm). Pink and green represent high and low proximity frequencies, respectively.”

3.11 - Figure 4E should be presented more explicitly and with appropriate statistics.

We have modified Figure 4E of the submitted manuscript by using a more conventional way to visualize the data. First, we now use the *intervene* tool from Khan and Mathelier (2017) to quantify the overlapping between Zelda and Class I Insulator peaks.

Next, we used *intersect* and *subtractBed* from bedtools v.2.30 to generate the corresponding .bed files (Quinlan and Hall 2010) for the different categories. The different bed files were processed using *computeMatrix* and plotted using *plotHeatmap* from deeptools 3.3.0 to create the heatmaps. Finally, we used a homemade matlab script to compute the average ATACseq signal profiles from the different groups of interest:

We observe that Class I IBPs sites overlapping with Zelda binding display an increase in accessibility. However we also noticed that Class I IBPs sites that are not overlapping with Zelda also display a significant increase in chromatin accessibility.

We changed the text as follows :

“This analysis revealed that only ~14% of the class I IBP sites corresponded to Zelda sites (Fig. 4j). Next, we calculated the accessibility of class I IBP sites at nc14 for all sites and for two subclasses: sites not bound by Zelda, and sites also bound by Zelda (Fig. 4k-l). Sites displaying both Class I IBPs and Zelda binding exhibited high accessibility, as expected. Notably, accessibility of Class I IBP sites not overlapping with Zelda represented the majority of sites and displayed significant accessibility. Overall, these results explain why preferential contacts between Class I insulators are not affected by Zelda depletion, and suggest that

this class of insulators rely on other means to access chromatin during early embryogenesis.”

REVIEWER COMMENTS

Reviewer #1 (Remarks to the Author):

The authors have addressed and responded to all my comments and inquiries in a thorough manner. I express my appreciation for the comprehensive efforts put forth by the authors. I believe that the manuscript has seen substantial improvement following its revision. Consequently, I recommend its publication.

Reviewer #2 (Remarks to the Author):

Thank you to the authors for the revised manuscript, which has improved clarity and largely addresses my previous comments. Overall, the data presented is largely convincing with respect to the conclusions of individual analyses. However, given the additional details about the Chromosight analysis I have two remaining concerns about the ChAS analysis, which are detailed below. In addition, the authors' findings of genome-wide enrichment of interactions between insulator-bound regions are somewhat in conflict with their findings at the *dpp* locus where regions that are bound by insulators colocalise only slightly more frequently than, or at similar frequencies to, those without insulators. In my opinion, the authors' model of insulators 'stabilising' infrequent interactions does not explain the discrepancy between the genome-wide Hi-C and locus-specific Hi-M results, and this discrepancy limits the generality of the conclusions that can be drawn from the data presented here. I would therefore suggest that the authors either amend the text to further describe how their model explains their results, or moderate the generalisation of the data and consider amending the title accordingly.

My specific concerns about the Chromosight/ChAS analyses:

1. From the additional details (numbers of loops called and overlap with other datasets) provided in the revised version, it's clear that Chromosight is calling substantially more 'loops' than previous analyses of *Drosophila* Hi-C/Micro-C data have identified. Consistent with this, the authors note in their response that a large proportion of the interactions do not appear to be focal loops, which is also clear in the new Fig S1 d. It appears likely that the majority of enriched interactions identified by Chromosight do not have the characteristics that most readers would expect from loops. Therefore, I would suggest that the authors clarify this in the main text and avoid/minimise the use of the term 'loop' to describe them.
2. The authors suggest that the low assortativity score for GAF, in contrast to previous results that identified GAF-bound loops, is due to the analysis being carried out "at *nc14*, before the midblastula transition". However, both Ogiyama et al. 2018 and Batut et al. 2022 identify GAF-associated loops at *nc14* (which is typically described as being at the end of the MBT, not before – Ogiyama et al. indeed describe these loops as appearing after the MBT). I believe an alternative explanation could be that such GAF-bound focal loops make up only a small proportion of the network analysed here, as seems likely based on the low fraction (~11%) of Chromosight-detected interaction anchors that overlap with the focal loop anchors from Batut et al. ChAS analysis using a more stringent set of loop calls, with a higher representation of focal loops, might therefore substantially change the assortativity score for GAF (and other factors). The authors could repeat the ChAS analysis using a more stringent set of loop calls, in addition to the more relaxed parameter sets they have now analysed, to confirm whether this is the case. If this analysis is not carried out, such alternative explanations for the discrepancy with existing literature should at least be discussed in the text.

Reviewer #3 (Remarks to the Author):

I thank the authors for the effort put into addressing the referees' comments and I am satisfied with most of the clarifications and additions that improved the manuscript's clarity and robustness.

However, **I still have some specific comments for the authors that need to be considered:**

- I strongly recommend that the authors carefully consider the terms "long-range" and "short-range" throughout the manuscript.

Mainly, I found misleading the title of the section:

"Insulator binding increases the strength of preferential long-range chromatin contacts" (line 132).

In this section, the main conclusions are drawn based on evidence reported at <250kb that are more short distances than long ones. Indeed, rightly, the author themselves state (lines 147-148):

"it can be inferred that Class I insulator sites exhibit a higher tendency to interact with each other at shorter distances".

This aspect needs to be clarified to improve the comprehension of the findings reported.

- I thank the author for performing the complementary analysis with two other sets of Chromosight parameters for loop calling that also account for lower-frequency interactions, as this is a crucial aspect for assessing the robustness of the computational method.

To have (to some extent) a quantifiable view of the long-range and low-frequencies interactions in the different networks, it will be informative to show how the distribution of the loop sizes changes in networks #2 and #3 compared to network #1.

I agree with the author that the complementary analysis supports that the overall trend in ChAs Z-scores is maintained even when considering long-range and low-frequencies interactions. However, the ChAs Z-scores seem highly network-specific (by numerical values). The author should comment on this aspect to improve interpretation by the readers. Indeed, with this recent analysis, some IBPs (CP190 and GAF) show considerably high ChAs Z-scores (>2) in network #3, which is the more permissive network. Does this suggest a long-range looping effect for these IBPs masked by the previous network construction?

This interesting aspect needs to be clarified further in the manuscript, especially considering that GAF has a role in forming repressive loops that are usually long-range interactions and considering the important remarks about GAF made previously by the other reviewers.

To this end, I recommend the author to re-modulate the sentence in line 129:

"GAF is not associated with chromatin loops."

And further comment on the interpretation of these results.

I also recommend that the authors add a reference line in Fig S1 that delineates the chosen ChAs Z-scores cut-off (set as 2) and add a better legend to label all the IBPs to improve clarity.

Additionally, I suggest a summary table to better recapitulate the ChAs Z-scores and the Log₂ O/E (in addition to Figure 1C) analysis of each IBP in all the considered networks can help facilitate a fair comparison. Especially since in the main text, the authors often mention different sets of IBPs at a given time and it is quite challenging to follow.

- I thank the author for including the Insulation Score analysis on the HiM-derived matrices. However, I still wonder how the HiM Insulation profile compares to the HiC Insulation profile, as the border and the TADs identified by the Insulation Score analysis seem weak. Indeed, as highlighted by the provided Domainogram, one of the TAD appears much more insulated than the first two. Is this trend also visible in the HiC Insulation profile? Is there a difference in insulators

protein within the strong and weak border? The author should comment on this aspect.

- The choice of cut-off in S3H (set at 200nm) is based on correlation with the HiC experiment that, by experiment design, is very good at detecting close-proximity interactions. Based on Figure S3H also a larger cut-off of 400nm still shows a good correlation with HiC of ~ 0.9 . This choice could mask longer multi-way interactions that occur significantly. Indeed, with a cut-off of 400nm the percentage of 3-way and 4-way contacts is conversably higher (20% and 10%, respectively; Figure S3H). The authors should comment on this and fully exploit the ability of HiM to investigate this further. I wonder how the trend observed in Figure 3D changes if larger cut-offs are considered.

- Regarding Figure S3I, I thank the author for including a set of 5 controls in the analysis. The authors report only the average values, what is the variation observed in the 5 controls? It will strengthen the author's main message to show that the insulator bin curve (in green) is within the variation found in the control.

- I found the conclusion of the section "Insulator-bound chromatin regions only infrequently co-localize in 3D" (lines 271-275) not fully supported by the results presented in this section and too speculative. I strongly suggest tuning down this conclusion and further commenting on this interesting interpretation in the discussion section.

Minor points:

Generally, I recommend carefully revisiting the figure legends and describing the panel's content in more detail.

- Line 146 "(referred to as peak 2 in Fig. S1h)", reference is not correct as S1H is the Heat map representing the ChAS Z-Scores from AND-ChAs analysis on the nc14 chromatin network. Please revise.

- Figure S1j, what is the color legend for the different IBPs?

- Figure 3D "score" overlaps one axis but is not the label of any axis.

- The method section "Multi-way proximity frequency analysis" (lines 547-554) does not mention which proximity frequency matrix has been used to generate Figure S3K.

- In Figure S3I, the authors show a bootstrap analysis to test that the number of traces acquired was sufficient to ensure a statistically representative ensemble map. Is the ensemble map being the HiC-derived matrix? This needs further clarification.

- In Figure 3D, unfortunately, the fitted curves are not too visible. Please update for clarity.

Point-by-point answer to reviewer's comments for:

3D chromatin interactions involving *Drosophila* insulators are infrequent but preferential and arise before TADs and transcription

Olivier Messina¹, Flavien Raynal², Julian Gurgo¹, Jean-Bernard Fiche¹, Vera Pancaldi^{2,3*}, Marcelo Nollmann^{1,*}

Affiliations

¹ Centre de Biologie Structurale, Univ Montpellier, CNRS UMR 5048, INSERM U1054, 34090 Montpellier, France.

² Université de Toulouse, Inserm, CNRS, Université Toulouse III-Paul Sabatier, Centre de Recherches en Cancérologie de Toulouse, 31037 Toulouse, France.

³ Barcelona Supercomputing Center, Barcelona, Spain.

* Corresponding authors: vera.pancaldi@inserm.fr, marcelo.nollmann@cbs.cnrs.fr

Color convention:

- *Reviewers comments are shown in gray italics*
- Our answers are provided in black.
- Changes to the revised manuscript are shown in blue.

Reviewer #1 (Remarks to the Author):

The authors have addressed and responded to all my comments and inquiries in a thorough manner. I express my appreciation for the comprehensive efforts put forth by the authors. I believe that the manuscript has seen substantial improvement following its revision. Consequently, I recommend its publication.

We thank the reviewer for the careful revision and positive comments.

Reviewer #2 (Remarks to the Author):

*2.1 Thank you to the authors for the revised manuscript, which has improved clarity and largely addresses my previous comments. Overall, the data presented is largely convincing with respect to the conclusions of individual analyses. However, given the additional details about the Chromosight analysis I have two remaining concerns about the ChAS analysis, which are detailed below. In addition, the authors' findings of genome-wide enrichment of interactions between insulator-bound regions are somewhat in conflict with their findings at the *dpp* locus where regions that are bound by insulators colocalise only slightly more frequently than, or at similar frequencies to, those without insulators. In my opinion, the authors' model of insulators 'stabilising' infrequent interactions does not explain the discrepancy between the genome-wide Hi-C and locus-specific Hi-M results, and this discrepancy limits the generality of the conclusions that can be drawn from the data presented here. I would therefore suggest that the authors either amend the text to further describe how their model explains their results, or moderate the generalisation of the data and consider amending the title accordingly.*

We thank the reviewer for this comment that indicates that our manuscript did not appropriately explain the model and how it is supported by both the HiM and the HiC data.

We understand that, at face value, the small insulator specificity in HiM results and the existence of preferential interactions in HiC seem contradictory. In fact, we think they are not, and we have performed three additional analyses to more clearly explain our interpretation of the HiC data and how it fits into the model.

First, we produced randomized snapshots of the distance normalized HiC data for regions that were selected by Chromosight as preferential loops (**Fig. S1d, below**). As this panel shows, many preferential Chromosight loops do not appear as clearly defined "focal peaks" usually seen, for instance, for CTCF loops in mammalian genomes. In fact, Chromosight loops display a more limited preference with respect to neighboring genomic regions, which makes them less well defined and with lower contrast. This is consistent with our HiM data at the *dpp* locus where we also do not see prominent "focal peaks" between insulator binding regions and where interactions between insulator barcodes display a limited preference. Thus, the genome-wide HiC analysis and our HiM data at the *dpp* locus are in fact consistent with each other and paint a similar picture.

Fig. S1d. Normalized maps for a randomized set of Chromosight loops

Second, we extended our $\log_2(O/E)$ and pile-up analyses to chart how preferential interactions between insulator binding regions change with the number of peaks being averaged (**Fig. S1o**, left panel). For this, we focused on BEAF-32, the insulator displaying the highest $\log_2(O/E)$ ratio and ChAs Z-score. When a small number of regions were averaged (<5), we observed a large dispersion of $\log_2(O/E)$ values, with many of them being negative (i.e. no preference). As expected, the average $\log_2(O/E)$ values (white circles in Fig. S1o) increased with the number of regions averaged, and plateaued after ~25. The dispersion in the $\log_2(O/E)$ values diminished with the number of regions averaged, as expected. Overall, these results are consistent with interactions between BEAF-32 anchors being highly variable and often displaying low or no preference. These results agree with our HiM data at the *dpp* locus where we observed only slight preferential interactions between insulator (mainly BEAF-32) bound sites.

Third, we performed Hi-C aggregate plot analysis with different numbers of BEAF-32 regions being piled-up (**Fig. S1o**, right panel). For each number of regions, we performed multiple averages using bootstrapping. As expected from the previous analysis, averaging of two regions displayed very variable pile-ups. Nonetheless, with a higher number of averaged regions (25 and over), the variability was reduced and the strength and contrast of the center pile-up peak was consistently increased. These analyses show that interactions between BEAF-32-bound regions display high heterogeneity and often exhibit weak or no preference, in agreement with our HiM data at the *dpp* locus.

Overall, we think these new analyses are fully consistent with our HiM results at the *dpp* locus, and support our model whereby interactions between insulator-bound regions are preferential *on average* but display high variability and often exhibit weak preference.

Fig. S1o. Variations in interaction preference with the number of regions averaged

We revised the Results section to limit the use of loops, to clarify that the preferential interactions we observe most often do not appear as focal peaks, and to introduce the new analyses presented above:

“...We applied ChAs analysis to study chromatin organization of *Drosophila* embryos at nuclear cycle 14 (nc14)²⁰, a developmental stage coinciding with the zygotic genome activation (ZGA) and with the emergence of TADs²⁰. For this, we obtained chromatin interaction networks by mapping preferentially interacting chromatin regions using Chromosight²⁹ on Hi-C data (**Fig. S1a-d**). Remarkably, the constructed network exhibits high overlap with previously annotated loops in the *Drosophila* embryo (**Fig. S1a**)³⁰. Chromosight detects preferential chromatin interactions by segmenting the genomic regions displaying local maxima in the observed/expected Hi-C map. In mammals, loops often appear as clear focal peaks⁷, however most of the Chromosight-annotated interactions from nc14 HiC data do not appear as focal peaks in the observed HiC map (**Fig. S1d**). This is consistent with many preferential contacts in *Drosophila* representing low-frequency interactions. Next, we annotated these chromatin networks with the binding patterns of publicly available ChIP-seq datasets (features, **Fig. 1a**) and calculated chromatin assortativities for a wide panel of chromatin binding factors, including insulator and insulator-associated proteins (BEAF-32, CBP, CHRO, CP190, dCTCF, DREF, FS(1)h, GAF, L(3)MBT, Pita, Mod(mdg4), Su(HW), Z4, ZIPIC and Zw5), pioneering factors (Zelda), RNA polymerase II (RNAPII CTD phospho-Ser5 : S5P), Polycomb group proteins (Pc, Ph) and the cohesin subunit (Rad21).”

“Preferential interactions captured by Chromosight are highly variable and often do not appear as focal peaks (Fig. S1d). We further analyzed the impact of this variability in our analysis by focusing on BEAF-32 –the insulator displaying the highest log₂(O/E) ratio and ChAs Z-score– and investigated how the interaction preference depended on the number of peaks aggregated. For this, we first calculated the distribution of log₂(O/E) values for different numbers of BEAF-32 peaks averaged using bootstrapping (**Fig. S1o, left panel**, see Log₂(O/E) and Hi-C aggregate plot analysis in Methods). On average, most of the 2- and 5-peak aggregations displayed low or no preference. Nonetheless, most aggregations exhibited positive log(O/E) values when 25 or more BEAF-32-bound regions were averaged. Overall, these results indicate that interactions between different BEAF-32 anchors are highly variable and often display low or no preference. In support of these conclusions,

well-centered peaks in Hi-C aggregate analysis were observed only after a sufficient number of BEAF-32-bound regions were aggregated (Fig. S1o, right panel). All in all, these analyses agree with our previous observations (Fig. S1d), and suggest that interactions between insulator-bound genomic regions are on average preferential, but highly variable and often weak.”

In addition, we revised the Discussion section to better explain the model and how it is supported by both our HiC analysis and HiM data.

“...Despite the genome-wide enrichment of IBPs at regions displaying 3D preferential interactions, the quantification of absolute proximity frequencies using Hi-M shows that insulator-bound regions (borders and non-borders) physically co-localize in space infrequently (~12%), and marginally more frequently than neighboring genomic regions (~9%). This observation is consistent with low proximity frequencies between TAD borders measured in S2 cells (~10%)³⁷. The low proximity frequencies between insulator-enriched regions are consistent with a recent study showing that depletion of insulators only partially weakens the strength of TAD borders⁴⁵, and with the overall absence of “focal loops” involving class I insulators in Hi-C contact maps^{4,6,20,31,56}. Finally, our genome-wide analysis shows that interactions between insulator-bound regions are on average preferential, but highly variable and often weak.”

“Previous studies proposed a role for *Drosophila* IBPs in mediating distant interactions^{13–15,62}. Our genome-wide analysis and imaging data are inconsistent with stable interactions between class I IBPs, and suggest that these insulators may play a role at stabilizing 3D distant chromatin conformations arising mainly from other processes, including polymer dynamics^{63,64}. It is well established that binding peaks from multiple insulators often cluster together^{16,49}. In this scenario, combinatorial binding of multiple insulator binding sites at single genomic locations^{19,45} would provide a means to modulate the strength of the stabilization, to regulate its specificity, and to enable a locus to time-share 3D interactions with multiple genomic locations in an asynchronous manner. Consistent with this concept, analyzing binding of RNAPII and polycomb members in mouse embryonic stem cell promoter-centered chromatin interactions using network measures such as bridgeness and betweenness centrality, it was suggested that RNAPII-bound chromatin fragments would belong to multiple communities at once, whereas polycomb bound fragments appeared to participate in multiple interactions at once²⁶.”

“...Direct measurements of residence times have, unfortunately, not been reported for class I *Drosophila* insulators. However, recent studies showed that GAF and mammalian CTCF can remain bound to their cognate chromatin sites for minutes^{66,67}, and that CTCF loops are dynamic^{63,68}. These data are consistent with a model whereby insulators help modulate the dynamics of specific interactions between distant cis-regulatory regions, but do not form stable scaffolds. These transient structures, however, may be more stable than the typical residence time of transcription factors (~10 seconds)⁶⁹. In this picture, insulators could help promote transcription by stabilizing transient cis-regulatory interactions to allow for the rapid binding and unbinding of transcription factors, or rather contribute to transcriptional repression by promoting 3D conformations that prevent functional interactions. This said, the lack of clear focal peaks, the high variability in interaction strength genome-wide, and the low proximity frequencies between class I insulator-bound regions, argue for the involvement of additional molecular actors in the 3D regulation of transcription.”

My specific concerns about the Chromosight/ChAS analyses:

2.2 From the additional details (numbers of loops called and overlap with other datasets) provided in the revised version, it's clear that Chromosight is calling substantially more 'loops' than previous analyses of Drosophila Hi-C/Micro-C data have identified. Consistent with this, the authors note in their response that a large proportion of the interactions do not appear to be focal loops, which is also clear in the new Fig S1 d. It appears likely that the majority of enriched interactions identified by Chromosight do not have the characteristics that most readers would expect from loops. Therefore, I would suggest that the authors clarify this in the main text and avoid/minimise the use of the term 'loop' to describe them.

The reviewer is correct, many loops called by Chromosight do not have the same characteristics as focal peaks in mammals. We have now underlined this clearly the first time we discussed loops called by Chromosight in the revised text. In addition, we updated our Supplementary Figure panel S1b to show a larger number of loops called by Chromosight. Finally, we minimized as much as possible the use of 'loop' in the text, as suggested by the reviewer.

The adapted text now reads:

"For this, we obtained chromatin interaction networks by mapping preferentially interacting chromatin regions using Chromosight ²⁹ on Hi-C data (**Fig. S1a-d**). Remarkably, the constructed network exhibits high overlap with previously annotated loops in the Drosophila embryo (**Fig. S1a**) ³⁰. Chromosight detects preferential chromatin interactions by segmenting the genomic regions displaying local maxima in the observed/expected Hi-C map. In mammals, loops often appear as clear focal peaks⁷, however most of the Chromosight-annotated interactions from nc14 HiC data do not appear as focal peaks in the distance normalized HiC map (**Fig. S1d**)."

2.3. The authors suggest that the low assortativity score for GAF, in contrast to previous results that identified GAF-bound loops, is due to the analysis being carried out "at nc14, before the midblastula transition". However, both Ogiyama et al. 2018 and Batut et al. 2022 identify GAF-associated loops at nc14 (which is typically described as being at the end of the MBT, not before – Ogiyama et al. indeed describe these loops as appearing after the MBT).

The reviewer is correct, nc14 corresponds to the mid-blastula transition. We have rephrased the text accordingly.

I believe an alternative explanation could be that such GAF-bound focal loops make up only a small proportion of the network analysed here, as seems likely based on the low fraction (~11%) of Chromosight-detected interaction anchors that overlap with the focal loop anchors from Batut et al. ChAS analysis using a more stringent set of loop calls, with a higher representation of focal loops, might therefore substantially change the assortativity score for GAF (and other factors). The authors could repeat the ChAS analysis using a more stringent set of loop calls, in addition to the more relaxed parameter sets they have now analysed, to confirm whether this is the case. If this analysis is not carried out, such alternative

explanations for the discrepancy with existing literature should at least be discussed in the text.

We agree with the reviewer in that an alternative explanation for the low-assortativity of GAF is that GAF-bound focal loops make up only a small proportion of the network. As the reviewer suggests, we now discuss this alternative explanation explicitly in the revised text:

“Low assortativity scores can arise when the presence of a factor is not associated with a preferential interaction (**Fig. 1a**), or if the factor is present either in a very small or in a very large proportion of them. For instance, GAF is often bound to the anchors of focal loops clearly visible in Hi-C and micro-C datasets^{30,31,33}. These focal loops, however, represent a small proportion of preferential interactions in our network (~11%, **Fig. S1a**), consistent with the low ChAs Z-scores we observed. We note that GAF binds to thousands of sites genome-wide (3842), however only a small fraction of these sites correspond to focal loop anchors (<620)³⁰. Taken together, these results are consistent with only a small number of GAF binding peaks being involved in focal loops and in regulating transcriptional activation and repression^{30,31,33}.”

Reviewer #3 (Remarks to the Author):

I thank the authors for the effort put into addressing the referees' comments and I am satisfied with most of the clarifications and additions that improved the manuscript's clarity and robustness.

However, I still have some specific comments for the authors that need to be considered:

3.1 *I strongly recommend that the authors carefully consider the terms "long-range" and "short-range" throughout the manuscript. Mainly, I found misleading the title of the section:*

"Insulator binding increases the strength of preferential long-range chromatin contacts" (line 132).

We apologize for the misleading title. The origin of this is that regulatory interactions in *Drosophila* -e.g. between enhancers and promoters- tend to be much shorter range (~ few kb) than in mammalian genomes (tens of kb). Also TADs in *Drosophila* are considerably smaller than in mammals. Thus, hundreds of kb is typically considered 'long-range' in *Drosophila*, while this would be considered short-range in other systems. This said, we agree that there is a confusion that needs to be addressed.

We addressed this issue by removing the mention of long-range from the title of this and other sections. We also revised the text to address this same issue in other sections (see answer 3.2).

3.2 *In this section, the main conclusions are drawn based on evidence reported at <250kb that are more short distances than long ones. Indeed, rightly, the author themselves state (lines 147-148):*

"it can be inferred that Class I insulator sites exhibit a higher tendency to interact with each other at shorter distances".

This aspect needs to be clarified to improve the comprehension of the findings reported.

We removed almost every use of 'long-range' in the manuscript to clarify this issue - including the offending sentence. This change does not affect the validity of the conclusions and avoids mis-interpretation. We thank the reviewer for this comment that will help clarify the message.

3.3 - *I thank the author for performing the complementary analysis with two other sets of ChromSight parameters for loop calling that also account for lower-frequency interactions, as this is a crucial aspect for assessing the robustness of the computational method.*

To have (to some extent) a quantifiable view of the long-range and low-frequencies interactions in the different networks, it will be informative to show how the distribution of the loop sizes changes in networks #2 and #3 compared to network #1.

We addressed this remark by analyzing the loop size distributions for networks 2 and 3. These new data are presented in Fig. S1h (see below). Both networks 2 and 3 encompass larger loops than network 1.

3.4 - I agree with the author that the complementary analysis supports that the overall trend in ChAs Z-scores is maintained even when considering long-range and low-frequencies interactions. However, the ChAs Z-scores seem highly network-specific (by numerical values). The author should comment on this aspect to improve interpretation by the readers. Indeed, with this recent analysis, some IBPs (CP190 and GAF) show considerably high ChAs Z-scores (>2) in network #3, which is the more permissive network. Does this suggest a long-range looping effect for these IBPs masked by the previous network construction? This interesting aspect needs to be clarified further in the manuscript, especially considering that GAF has a role in forming repressive loops that are usually long-range interactions and considering the important remarks about GAF made previously by the other reviewers.

We thank the reviewer for this comment to help further clarify our results regarding GAF.

As **Fig. S1h** shows, networks 2 and 3 both capture a broader range of loop sizes than network 1, with network 3 capturing the longest loops. The ChAs Z-scores for most insulators are highly correlated between networks (**Fig. S1g**) and tend to increase with the average loop size of the network. As the reviewer, we remark that for some insulators the ChAs Z-score increase is larger than proportional in the networks including longer-range contacts (e.g. GAF), while for others the ChAs Z-score increased less than proportionally (e.g. Fs1h, CTCF). This indicates that GAF is slightly more assortative in networks with longer-range loops, while FS1h/CTCF are less assortative in this network. Overall, however, these factors still display lower assortativities than class I IBPs for all networks considered.

We now discuss this in the Results section (see below).

To this end, I recommend the author to re-modulate the sentence in line 129:

*"GAF is not associated with chromatin loops."
And further comment on the interpretation of these results.*

We considerably revised the paragraph containing the offending sentence (see below). In addition, we also provide alternative explanations for the relatively low assortativity of GAF,

which is always smaller than that of other insulators irrespective of the network used for the analysis.

The two revised paragraphs of the Results section now read:

“Chromatin assortativity Z-scores (hereafter ChAs Z-scores) are calculated to estimate if ChAs for a feature is higher than expected for regions separated by similar genomic distances, indicating the importance of 3D interactions for establishing preferential contacts. Regions enriched in Zelda, Polycomb group proteins (Pc and Ph), and RNAPII CTD phospho-Ser5 (S5P) displayed positive ChAs Z-scores (**Fig. S1e**), consistent with previous findings^{31–33}. In contrast, ChAs Z-scores were highly variable between IBPs (**Fig. 1b**), indicating that different insulators may contribute unequally to the formation of preferential contacts. A sub-group of IBPs displayed high assortativities (ChAs Z-score > 2), including the insulator and insulator-associated proteins: BEAF-32, CHRO, DREF, L(3)MBT, Pita, Z4, ZIPIC and Zw5 (**Fig. 1b**). Notably, cohesin (Rad21), dCTCF, and a second sub-group of IBPs including CBP, CP190, Fs(1)h, GAF, Mod(mdg4) and SU(HW) displayed low assortativity and low Z-scores (ChAs Z-score < 2, **Figs. 1b, S1e**). To validate the robustness of these results, we performed similar analysis for different sets of Chromosight parameters (see *chromatin assortativity* in Methods) generating larger networks that include lower-frequency interactions. ChAs Z-scores were highly correlated between networks, and the insulator factors exhibiting the highest ChAs Z-scores were the same independently of the network size or loop size distribution (**Figs. S1f-h**). For some insulators the ChAs Z-score increase was larger than proportional in the networks including longer-range contacts (e.g. GAF), while for others the ChAs Z-score increased less than proportionally (e.g. Fs1h, CTCF). This is consistent with these factors being slightly more/less assortative depending on the network loop size distribution. We note, however, that these factors still displayed the lowest assortativities in all networks.”

“Low assortativity scores can arise when the presence of a factor is not associated with a preferential interaction (**Fig. 1a**), or if the factor is present either in a very small or in a very large proportion of them. For instance, GAF is often bound to the anchors of focal loops clearly visible in Hi-C and micro-C datasets^{30,31,33}. These focal loops, however, represent a small proportion of preferential interactions in our network (~11%, **Fig. S1a**), consistent with the low ChAs Z-scores we observed. We note that GAF binds to thousands of sites genome-wide (3842), however only a small fraction of these sites correspond to focal loop anchors (<620)³⁰. Taken together, these results are consistent with only a small number of GAF binding peaks being involved in focal loops and in regulating transcriptional activation and repression^{30,31,33}.”

3.5 I also recommend that the authors add a reference line in Fig S1 that delineates the chosen ChAs Z-scores cut-off (set as 2) and add a better legend to label all the IBPs to improve clarity.

We have adapted the relevant figure panels (Figs. S1f and S1g) to add the ChAs Z-score cut-off as recommended, and also increased the size of the IBP labels to improve clarity.

3.6 - Additionally, I suggest a summary table to better recapitulate the ChAs Z-scores and the Log2 O/E (in addition to Figure 1C) analysis of each IBP in all the considered networks can help facilitate a fair comparison. Especially since in the main text, the authors often mention different sets of IBPs at a given time and it is quite challenging to follow.

We have now included a Supplementary table summarizing the ChAs, and log2(O/E) results:

Supplementary Table 4															
	BEAF-32	CBP	CHRO	CP190	dCTCF	DREF	Fs(1)h	GAF	L(3)MBT	Mod(mdg4)	Pita	SU(HW)	Z4	ZIPIC	Zw5
Chas-Z-Score	9.3	0.3	7.7	1.6	1.7	4	1.3	0.2	4.3	0.7	2.7	0.7	8.4	2.3	3.3
mean of log2(O/E)	0.23	-0.16	0.16	-0.04	-0.21	0.21	-0.12	-0.21	-0.02	-0.18	-0.08	-0.50	0.14	0.11	0.06
Class	I	II	I	II	II	I	II	II	II	II	II	II	I	I	I

3.7 - I thank the author for including the Insulation Score analysis on the HiM-derived matrices. However, I still wonder how the HiM Insulation profile compares to the HiC Insulation profile, as the border and the TADs identified by the Insulation Score analysis seem weak. Indeed, as highlighted by the provided Domainogram, one of the TAD appears much more insulated than the first two. Is this trend also visible in the HiC Insulation profile? Is there a difference in insulators protein within the strong and weak border? The author should comment on this aspect.

To address these additional questions of the reviewer, we calculated the domainogram for the HiC dataset used in the manuscript (Hug, *et al*, 2017). The domainogram analysis from HiM and HiC data at the *dpp* locus are remarkably similar (**Fig. R1**, below). As expected, the last TAD at the *dpp* locus appears more insulated than the first two in both analyses. While the borders between TAD1 and TAD2 do not display insulator binding, the borders of TAD3

display several binding peaks (~2-3) for multiple IBPs (see Fig. 3b). This is consistent with the role of IBPs in TAD insulation in *Drosophila*.

We modified the text to comment on this aspect as follows:

“The proximity and PWD distance maps revealed multiple regions displaying preferential 3D spatial proximity (**Fig. 3c**). These mostly corresponded to the TADs called from Hi-C data (Figs. 3b-c, blue arrows) and from Hi-M proximity frequency maps (Fig. 3c, insulation score, and domainogram). **The last TAD in this region is flanked by multiple IBP peaks, and appears considerably more insulated than the other TADs in this region, consistent with the role of IBPs in TAD insulation.**“

3.8 The choice of cut-off in S3H (set at 200nm) is based on correlation with the HiC experiment that, by experiment design, is very good at detecting close-proximity interactions. Based on Figure S3H also a larger cut-off of 400nm still shows a good correlation with HiC of ~0.9. This choice could mask longer multi-way interactions that occur significantly. Indeed, with a cut-off of 400nm the percentage of 3-way and 4-way contacts is conversably higher (20% and 10%, respectively; Figure S3H). The authors should comment on this and fully exploit the ability of HiM to investigate this further.

We agree with the fact that other distance thresholds (e.g. 400nm) produce HiM matrices that are still in very good agreement with HiC maps. However, we are worried that considering such large cutoff distances may not be compatible with the accepted definition of multiway contacts.

Multi-way contacts are typically defined as the spatial clustering of genomically-distant regions. Because of the relatively small genomic size of the *dpp* locus, most barcodes at this locus have a high probability of finding each other for large proximity distances (e.g. 400nm) (see pairwise distance map in Fig. 3c). However, we don't think this is necessarily indicative of formation of multiway contacts between specific genomic elements, as one would obtain similar results for a random-coil polymer. In other words, the fact that multiple barcodes occupy the same ~400nm volume does not necessarily mean they form specific multi-way contacts.

Increasing the cutoff distance will indeed increase the frequency of proximity (Fig. 3d), but this would happen for both insulator and non-insulator barcodes. We revised the text to note that the percentage of multiway contacts increases for large cutoff distances, and comment on our interpretation.

“Finally, to explore if these rare spatial encounters involved multiple insulator-bound regions, we calculated the proportion of clusters containing two (i.e. pairwise cluster) or multiple insulator barcodes (multiway cluster). Clusters containing only two insulator targets were the most common in all cases (>65%) (**Fig. 3f**). Next, we calculated the frequency of multi-way clusters as a function of the number of barcodes in a cluster for all barcodes combined (**Fig. 3f**) or for each barcode independently at a distance ≤ 200 nm (**Fig. 3g**) and for different distance thresholds (**Fig. S3k**). We note that at larger cutoff distances (e.g. 400 nm) multiple barcodes can frequently coalesce in space, but we don't consider these to represent multiway clusters because of the large distances involved. The frequency of multi-way clusters rapidly decreased with the number of co-localizing targets but was still slightly higher than what would be expected by chance (**Fig. 3f**, see *Multiway proximity frequency analysis Methods*).”

I wonder how the trend observed in Figure 3D changes if larger cut-offs are considered.

Proximity frequencies increase with the distance cutoff (Fig. S3g), however, these frequencies would increase for both insulator and control barcodes. We revised the text to comment on this aspect:

“Proximity frequencies dropped with genomic distance, as expected, but the difference between insulator and non-insulator barcodes remained small for all genomic distances. We note that use of larger cutoff distances increases the proximity frequency, but this would happen for both insulator and non-insulator barcodes.”

3.9 - *Regarding Figure S3I, I thank the author for including a set of 5 controls in the analysis. The authors report only the average values, what is the variation observed in the 5 controls? It will strengthen the author's main message to show that the insulator bin curve (in green) is within the variation found in the control.*

To address this comment, we calculated the standard deviation for the controls used in the analysis in the revised Fig. S3j, which now replaces the original inset. We also revised the original panel legend as follows:

“j. Cumulative proximity frequency versus different cutoff distances for Class I IBP barcodes (green) and for 10 sets of control barcodes (black) for nc14 embryos. For the control, the solid black line represents the mean and the gray shade represents two standard deviations calculated from the variability of controls.”

Fig. S3j. The average for the control regions (black) is shown together with its standard deviation in the inset (gray shade).

3.10 - I found the conclusion of the section "Insulator-bound chromatin regions only infrequently co-localize in 3D" (lines 271-275) not fully supported by the results presented in this section and too speculative. I strongly suggest tuning down this conclusion and further commenting on this interesting interpretation in the discussion section.

As recommended by the reviewer, we removed the conclusion for this section, as all points are already discussed in the Discussion section. This section now ends by:

"Overall, these results show that insulators coalesce in space infrequently, and only at slightly higher frequencies than non-insulator regions."

Minor points:

Generally, I recommend carefully revisiting the figure legends and describing the panel's content in more detail.

3.11 - Line 146 "(referred to as peak 2 in Fig. S1h)", reference is not correct as S1H is the Heat map representing the ChAS Z-Scores from AND-ChAs analysis on the nc14 chromatin network. Please revise.

We thank the reviewer for pointing this out, and have corrected the text accordingly:

"The peaks observed for negative $\log_2(O/E)$ values (referred to as peak 2 in Fig. S1j)."

3.12 Figure S1j, what is the color legend for the different IBPs?

We edited the legend of S1j to specify the colors used for the different IBPs, as follows:

“j. Violin plots illustrating the distribution of the $\log_2(O/E)$ for 15 IBPs, **green**, the cohesin subunit (Rad21), **pink**, Polycomb group proteins (Pc, Ph), **blue**, RNA polymerase II (RNAPII CTD phospho-Ser5) and the pioneering factors (Zelda), **red**, in the nc14 chromatin network for all genomic distances classified by alphabetical order. Dashed rectangles pinpoint the two different peaks observed in the distribution. k. For genomic distances shorter than 250 kb.”

3.13 Figure 3D "score" overlaps one axis but is not the label of any axis.

We corrected the figure to address this issue.

3.14 The method section "Multi-way proximity frequency analysis" (lines 547-554) does not mention which proximity frequency matrix has been used to generate Figure S3K.

The legend of Fig S3k indicated that this analysis was performed on the nc14 matrix. We now provide this information also in the methods, as follows :

The proportion of multiway contacts is calculated from single nucleus proximity frequency **nc14** matrices as described previously⁷³.

3.15 In Figure S3l, the authors show a bootstrap analysis to test that the number of traces acquired was sufficient to ensure a statistically representative ensemble map. Is the ensemble map being the HiC-derived matrix? This needs further clarification.

We apologize for the confusion. The ensemble map is the HiM map obtained by averaging the contacts of all the traces. We amended the legend to clarify, as follows:

Violin plots representing the Pearson correlation between and the **Hi-M** ensemble matrix and matrices generated by sampling subsets of traces by bootstrapping. **The Hi-M ensemble matrix was obtained by considering all the traces available.** For each condition, 250 bootstrapping cycles were used.

3.16 In Figure 3D, unfortunately, the fitted curves are not too visible. Please update for clarity.

We changed the color of the black curve to gray to make it more visible.

Revised panel Fig. 3d

REVIEWERS' COMMENTS

Reviewer #2 (Remarks to the Author):

I thank the authors for the effort put in to provide the additional explanations and analyses to clarify the results and their model. I am satisfied with the responses to my comments and therefore recommend the manuscript for publication.

Reviewer #3 (Remarks to the Author):

The authors have addressed my major critiques and inquiries. As a result of their detailed revision, the manuscript has undergone noteworthy improvement, and I recommend its publication.